# Learning to Embed Distributions via Maximum Kernel Entropy

**Oleksii Kachaiev**
Dipartimento di Matematica, Università degli Studi di Genova, Genoa, Italy
`oleksii.kachaiev@gmail.com`

**Stefano Recanatesi**
Technion Israel Institute of Technology, Haifa, Israel
Allen Institute for Neural Dynamics, Seattle, USA
`stefano.recanatesi@gmail.com`

## Abstract

Empirical data can often be considered as samples from a set of probability distributions. Kernel methods have emerged as a natural approach for learning to classify these distributions. Although numerous kernels between distributions have been proposed, applying kernel methods to distribution regression tasks remains challenging, primarily because selecting a suitable kernel is not straightforward. Surprisingly, the question of learning a data-dependent distribution kernel has received little attention. In this paper, we propose a novel objective for the unsupervised learning of data-dependent distribution kernel, based on the principle of entropy maximization in the space of probability measure embeddings. We examine the theoretical properties of the latent embedding space induced by our objective, demonstrating that its geometric structure is well-suited for solving downstream discriminative tasks. Finally, we demonstrate the performance of the learned kernel across different modalities.

## 1 Introduction

Most discriminative learning methods conventionally assume that each data point is represented as a real-valued vector. In practical scenarios, however, data points often manifest as a 'set' of features or a 'group' of objects. A quintessential example is the task of predicting a health indicator based on multiple blood measurements. In this case, the single data point of a patient has multiple, or a distribution of, measurements. One approach to accommodate such cases involves representing each input point as a probability distribution. Beyond mere convenience, it is more appropriate to model input points as distributions when dealing with missing data or measurement uncertainty, as often encountered when facing the abundance of data, which commonly presents a challenge for data-rich fields such as genetics, neuroscience, meteorology, astrophysics, or economics.

The task of regressing a mapping of probability distributions to a real-valued response is known as *distribution regression*. Distribution regression has been successfully applied in various fields, such as voting behavior prediction [13], dark matter halo mass learning [43], human cancer cells detection [44], brain-age prediction [6], among others [38, 33, 68]. The versatility and effectiveness of this framework underscore its power in solving complex problems [69, 29]. Kernel methods have become a widely used approach for solving distribution regression tasks by exploiting a kernel between distributions referred to as a *distribution kernel*. Despite the multitude of proposed kernels, the practical application of kernel methods remains challenging due to the nontrivial choice of the appropriate kernel. While some efforts focus on identifying kernels with broad applicability and

38th Conference on Neural Information Processing Systems (NeurIPS 2024).

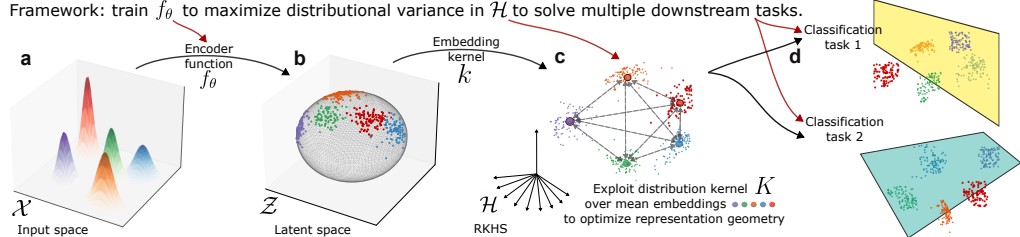

Figure 1: **Learning to embed distributions.** (a) Example of multiple distributions over the input space. (b) The trainable function $f_\theta$ encodes the input dataset into a compact latent space, in our case $\mathcal{Z} = \mathcal{S}^{d-1}$. (c) The first-level *embedding* kernel $k$ induces kernel mean embedding map to $\mathcal{H}$. The encoder is optimized to maximize the entropy of the covariance operator embedding of the dataset w.r.t. the second-level *distribution* kernel $K$ between kernel mean embeddings in $\mathcal{H}$. (d) Utilizing learned data-dependent kernel, downstream classification tasks can be solved using tools such as Kernel SVM or Kernel Ridge Regression.

favorable statistical properties [54], others aim to tailor kernels to the geometric characteristics of specific input spaces [6]. Remarkably, the question of learning data-dependent kernels has received limited attention. This study is thus driven by a fundamental question: What are the underlying principles that facilitate the unsupervised learning of an effective kernel, one that optimally encapsulates the data properties and is well suited for discriminative learning on distributions?

In this work, we leverage a key insight: an appropriate selection of the distribution kernel enables the embedding of a set of distributions into the space of covariance operators. Building on this theoretical idea, we claim that quantum entropy maximization of the corresponding covariance operator is a suitable guiding principle to learn data-dependent kernel. This, combined with a careful design of kernel parametrization, let us to devise a differentiable optimization objective for learning data-specific distribution embedding kernels from unlabeled datasets (i.e. unsupervised) (Fig. 1). We show that the entropy maximization principle facilitates learning of the latent space with geometrical configuration suitable for solving discriminative tasks [3, 20]. We empirically demonstrate the performance of our method by performing classification tasks in multiple modalities.

In summary, our unsupervised data-dependent distribution kernel learning framework introduces a theoretically grounded alternative to the common practice of hand-picking kernels. Such framework could be further leveraged for generalizing existing learning approaches [61, 32] catalyzing the use of distribution-based representations within the broader scientific community.

## 2 Preliminaries

We first introduce the main concepts necessary to formalize our learning framework: kernel mean embeddings and covariance operator embeddings.

### 2.1 Kernel Embeddings of Distributions

Consider an input space $\mathcal{X}$ and a positive-definite (p.d.) kernel $k : \mathcal{X} \times \mathcal{X} \to \mathbb{R}$. Let $\mathcal{H}$ be the corresponding reproducible kernel Hilbert space (RKHS) induced by such kernel. Consider a probability distribution $P \in \mathcal{P}(\mathcal{X})$. The *kernel mean embedding* map embeds the distribution $P$ as a function in Hilbert space:

$$\mu_P \equiv \mu(P) := \int_\mathcal{X} k(x, \cdot) \, dP(x) = \int_\mathcal{X} \phi(x) \, dP(x) \,, \tag{1}$$

where $\phi : \mathcal{X} \to \mathcal{H}$ is a feature map such that $\phi(x) = k(x, \cdot)$.

Importantly, if the kernel $k$ is *characteristic* [50], the mapping $\mu : \mathcal{P}(\mathcal{X}) \to \mathcal{H}$ is injective, implying that all information about the original distribution is preserved in $\mathcal{H}$. This last property underscores much of power under recent applications of kernel mean embeddings [41, 48, 14]. The natural empirical estimator for the kernel mean embedding approximates the true distribution with a finite

sum of Dirac delta functions:

$$\hat{\mu}_P := \frac{1}{N} \sum_{i=1}^{N} \phi(x_i) \in \mathcal{H} \tag{2}$$

where $x_1, \ldots, x_N \sim P$ are $N$ empirical i.i.d. samples. The estimator has a dimension-free sample complexity error rate of $\mathcal{O}(N^{-\frac{1}{2}})$ [51].

Additionally, we denote $d_k : \mathcal{X} \times \mathcal{X} \to \mathbb{R}_{\geq 0}$ a *kernel metric* induced by a kernel $k$,

$$d_k(x, x') = \|\phi(x) - \phi(x')\|_{\mathcal{H}}. \tag{3}$$

Note that $d_k$ is a metric on $\mathcal{X}$ if feature map $\phi$ is injective.

## 2.2 Covariance Operators and Entropy

A second way of mapping a probability distribution to a Hilbert space can be defined by means of a covariance operators. For a given feature map $\phi(x) = k(x, \cdot) : \mathcal{X} \to \mathcal{H}^1$, and a given probability distribution $P \in \mathcal{P}(\mathcal{X})$, the *covariance operator embedding* is defined as:

$$\Sigma_P := \int_{\mathcal{X}} \phi(x) \otimes \phi(x) \, dP(x) \tag{4}$$

where $\otimes$ is a tensor product. $\Sigma_P$ is a self-adjoint positive semi-definite (p.s.d.) operator acting on $\mathcal{H}$. Such operator can be seen as a mean embedding w.r.t. the feature map $x \mapsto \phi(x) \otimes \phi(x)$ and therefore, for a *universal* kernel $k$, the map $P \mapsto \Sigma_P$ is injective (see Bach [1]).

Similarly to Eq. (2), the natural empirical estimator is:

$$\hat{\Sigma}_P = \frac{1}{N} \sum_{i=1}^{N} \phi(x_i) \otimes \phi(x_i) \tag{5}$$

where $x_1, \ldots, x_N \sim P$ are $N$ i.i.d. samples.

For a translation-invariant kernel $k(x, x') = \psi(x - x')$ normalized such that $k(x, x) = 1$, the covariance operator $\Sigma_P$ is a *density operator* [1]. Henceforth, entropy measures can be applied to it, and the quantum Rényi entropy of the order $\alpha$ can be defined as:

$$\mathcal{S}_\alpha(\Sigma_P) := \frac{1}{1-\alpha} \log \operatorname{tr}\left[(\Sigma_P)^\alpha\right] = \frac{1}{1-\alpha} \log \sum_i \lambda_i^\alpha \tag{6}$$

where $\{\lambda_i\}_i$ are the eigenvalues of $\Sigma_P$. The Von Neumann entropy can be seen as a special case of Rényi entropy in the limit $\alpha \to 1$. However, in our work, we focus primarily on the second-order case of Rényi entropy, i.e. $\alpha = 2$ (see Carlen [9], Müller-Lennert et al. [42], Wilde [63], Giraldo et al. [15] for an in-depth overview of the properties and theory of quantum entropies).

# 3 Unsupervised Distribution Kernel Learning

## 3.1 Distribution Regression

In this section, we discuss the key topics in distribution regression, including problem setup, the notion of a 2-stage sampling process, and the common solutions to regression employing kernel methods.

*Distribution regression* extends the common regression framework to the setup where covariates are given as probability distributions available only through samples. Formally, consider the task of finding a regressor $f : \mathcal{P}(\mathcal{X}) \to \mathcal{Y}$ from the dataset of samples $\Gamma_M = \{(P_i, y_i)\}_{i=1}^M$ where $P_i \in \mathcal{P}(\mathcal{X})$ are distributions provided as a set of i.i.d. empirical samples $x_1, \ldots, x_{N_i} \sim P_i$ (see Poczos et al. [45], Szabó et al. [53, 54] for a comprehensive analysis). A viable approach to solving this problem is to define a kernel $K : \mathcal{P}(\mathcal{X}) \times \mathcal{P}(\mathcal{X}) \to \mathbb{R}$ that is *universal* in $\mathcal{P}(\mathcal{X})$. By setting up such a kernel $K$, we can utilize kernel-based regression techniques, with SVM often being the

---

[1] Assuming $k$ to be a continuous positive definite (p.d.) kernel and $\mathcal{X}$ to be compact.

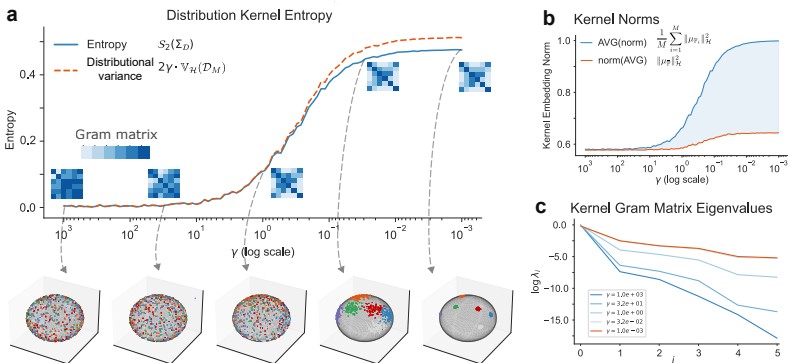

Figure 2: **Properties of the entropy on the toy example.** (a) Entropy and Distributional Variance for 6 distributions on a sphere as a function of their geometrical arrangement parametrized by $\gamma$. (b) Kernel norms that enter the distributional variance bound. The blue shaded area (difference between blue and red lines) corresponds to the dotted red line in (a) (up to multiplicative factor). (c) Flattening of Gram matrix eigenvalues as a function of $\gamma$.

preferred method for classification tasks [39] or Kernel Ridge Regression (KRR) when the space $\mathcal{Y}$ is continuous [37] [2]. To this end, several kernels have been proposed over time (see details in Sec. 4).

One possibility, proposed by Muandet et al. [39], is to introduce a kernel in the input space, the so called *embedding kernel* $k_{\text{emb}} : \mathcal{X} \times \mathcal{X} \to \mathbb{R}$ and exploit the induced mean embeddings $\mu_{\text{emb}} : \mathcal{P}(\mathcal{X}) \to \mathcal{H}_{\text{emb}}$ to map input distributions to points in RKHS. Subsequently, to define a second level kernel, *distribution kernel* $K_{\text{distr}} : \mathcal{H}_{\text{emb}} \times \mathcal{H}_{\text{emb}} \to \mathbb{R}$ between points in the RKHS $\mathcal{H}_{\text{emb}}$. The simplest choice for such a distribution kernel is the linear kernel:

$$K_l(\mu_P, \mu_Q) := \langle \mu_P, \mu_Q \rangle_{\mathcal{H}_{\text{emb}}} = \iint_{\mathcal{X} \times \mathcal{X}} k(x, x') \, dP(x) \, dQ(x') \,. \tag{7}$$

A standard alternative to the linear kernel is a Gaussian kernel with bandwidth parameter $\lambda > 0$:

$$K_{\text{RBF}}(\mu_P, \mu_Q) := \exp\left(-\frac{\lambda}{2}\|\mu_P - \mu_Q\|^2_{\mathcal{H}_{\text{emb}}}\right) = \exp\left(-\frac{\lambda}{2} d_{k_{\text{emb}}}(\mu_P, \mu_Q)^2\right), \tag{8}$$

which was shown to be *universal* in $\mathcal{P}(\mathcal{X})$ [11]. The Gaussian kernel $K_{\text{RBF}}$ can be computed from the linear kernel $K_l$ as $d_{k_{\text{emb}}}(\mu_P, \mu_Q)^2 = K_l(P, P) + K_l(Q, Q) - 2K_l(P, Q)$.

In practice, we only have access to a finite number of samples of each input distribution, thus the *distribution kernel* is approximated using the natural estimator for the kernel mean embedding. The related excess risk for the regression solution is analyzed in Szabó et al. [53].

### 3.2 Dataset Embedding

Instead of using standard kernels designed to encapsulate the geometry of the input space, we consider learning a data-dependent kernel, tailored to the specific properties of the dataset. In a similar vein, Yoshikawa et al. [68] proposed learning an optimal kernel (or equivalently, a feature map) jointly with the classifier to address the text modality. In this work, we focus on an *unsupervised* problem, aiming to learn a data-dependent kernel between probability distributions without access to classification labels.

We first introduce proper parametrization to ensure both expressivity and robustness followed by the definition of the optimization objective. Leveraging the idea of 2-level kernel setup, we define the *embedding kernel* as

$$k_\theta : \mathcal{X} \times \mathcal{X} \to \mathbb{R} = k_{\text{emb}}(f_\theta(x), f_\theta(x')) \,. \tag{9}$$

where $f_\theta$ is a trainable encoder function $f_\theta : \mathcal{X} \to \mathcal{Z}$, $\mathcal{Z}$ is a latent encoding space, and $k_{\text{emb}}$ is a kernel defined on the latent space $k_{\text{emb}} : \mathcal{Z} \times \mathcal{Z} \to \mathbb{R}$. The encoder function $f_\theta$ transforms every

---

[2]As pointed out in Meunier et al. [37], under mild conditions, KRR can also be used for classification problems.

input probability distribution $P \in \mathcal{P}(\mathcal{X})$ into a distribution over the latent space $\mathbb{P}_\theta \in \mathcal{P}(\mathcal{Z})$ [3] (Fig. 1a). Furthermore, we denote RKHS corresponding to the kernel $k_{\text{emb}}$ as $\mathcal{H}_{\text{emb}}$ and the kernel mean embedding map as $\mu_{\text{emb}}$ (see Eq. (1)).

$$P \in \mathcal{P}(\mathcal{X}) \xrightarrow{f_\theta} P_\theta \in \mathcal{P}(\mathcal{Z}) \xrightarrow{k_{\text{emb}}} \mu_{\text{emb}}(P) \in \mathcal{H}_{\text{emb}} . \tag{10}$$

These transformations define mean embeddings for each input probability distributions through the first level, embedding kernel $k_{\text{emb}}$ (Fig. 1b). The second level, *distribution kernel* $K_{\text{distr}}$ : $\mathcal{H}_{\text{emb}} \times \mathcal{H}_{\text{emb}} \to \mathbb{R}$ is defined over the mean embeddings $\mu_{\text{emb}}(P)$'s. We can now consider to embed dataset $\mathcal{D}_M = \{P_i\}_{i=1}^M$ as an empirical covariance operator (see Eq. (5)), i.e.

$$\mathcal{D}_M \xrightarrow{\mu_{\text{emb}}, K_{\text{distr}}} \hat{\Sigma}_\mathcal{D} = \frac{1}{M} \sum_{P \in \mathcal{D}_M} K_{\text{distr}}(\mu_{\text{emb}}(P), \cdot) \otimes K_{\text{distr}}(\mu_{\text{emb}}(P), \cdot) . \tag{11}$$

As $\hat{\Sigma}_\mathcal{D}$ encapsulates information about the entire dataset, we term it *dataset embedding*. With the assumption that the dataset $\mathcal{D}_M$ is sampled i.i.d. from the (unknown) true meta-distribution $\mathcal{D}$, $\hat{\Sigma}_\mathcal{D}$ is a natural estimator to approximate the true covariance operator $\Sigma_\mathcal{D}$. To simplify the notation we use $\Sigma_\mathcal{D}$ in place of $\Sigma_\mathcal{D}$ unless required by the context.

Both the *embedding* kernel $k_{\text{emb}}$ and the *distribution* kernel $K_{\text{distr}}$ remain fixed throughout the training, learning happens by adjusting the parametrization of the latent space encoder $f_\theta$. Such separation ensures expressivity while conforming to all technical requirements for a distribution kernel. Throughout the paper, we make the following assumptions on the latent space $\mathcal{Z}$ and embedding kernel $k$.

**Assumption 3.1.** Latent space $\mathcal{Z}$ is a compact subset of $\mathbb{R}^d$. Kernel $k_{\text{emb}} : \mathcal{Z} \times \mathcal{Z} \to \mathbb{R}$ is a p.d. characteristic translation-invariant kernel $k_{\text{emb}}(z, z') = f(\|z - z'\|^2)$ such that $-f'$ is completely monotone on $(0, \infty)$ (see Definition 2.2.4 of Borodachov et al. [7]) and $\forall z \in \mathcal{Z}: k_{\text{emb}}(z, z) = 1$.

The choice of a kernel based on Euclidean distance in the latent space makes its definition similar to that in Weinberger and Tesauro [62], though the parametrization of the encoding process differs. Learning kernels by explicitly learning feature maps has been explored in a wide range of settings [66, 57, 58, 67]. In contrast, the parametrization proposed in our study applies a known characteristic kernel to a learned latent representation. To facilitate our optimization process (which will be explained shortly), we opt for $\mathcal{Z} = \mathbb{S}^{d-1}$ (the d-dimensional hypersphere) and Gaussian kernel both for $k_{\text{emb}}$ and $K_{\text{distr}}$ (see Eq. (8)). We retain other suitable choices as potential avenues for future research.

## 3.3 Unsupervised Optimization Objective

This dataset level representation depends on the choice of first and second level kernels $k, K$ and, in turn, on the trainable function $f_\theta$ parameterized by the set of parameters $\theta$. In this work, we propose learning the parameters $\theta$ to **maximize quantum entropy** of the *dataset embedding*, i.e.,

$$\theta = \arg \max \left\{ \mathcal{S}_2(\Sigma_\mathcal{D}) := -\log \text{tr}\left[(\Sigma_\mathcal{D})^2\right] \right\} . \tag{12}$$

As we will describe in brief, optimizing this target has clear benefits inherited from the underlying geometry of the setup. But, first, we show how to empirically compute $\mathcal{S}_2(\Sigma_\mathcal{D})$. Building upon previous work [1][4], we exploit the following property of the covariance estimator:

$$\text{tr}\left[(\Sigma_\mathcal{D})^2\right] = \text{tr}\left[\left(\frac{1}{M} K_\mathcal{D}\right)^2\right] \tag{13}$$

where $K_\mathcal{D} \in \mathbb{R}^{M \times M}$ is the *distribution kernel* matrix, with $[K_\mathcal{D}]_{ij} = K_{\text{distr}}(\mu_{P_i}, \mu_{P_j})$. This equation follows directly from the fact that $\Sigma_\mathcal{D}$ and $\frac{1}{M} K_\mathcal{D}$ share the same set of eigenvalues. Leveraging this

---

[3]By the definition of the encoding process, $P_\theta$ is a push-forward measure. For an empirical probability distribution $q = \sum_i \delta(x_i) \in \mathcal{P}(\mathcal{X})$ and a measurable map $f : \mathcal{X} \to \mathcal{Z}$, the push-forward measure $f \# q \in \mathcal{P}(\mathcal{Z})$ is defined as $\sum_i \delta(f(x_i))$.

[4]See the proof of Proposition 6 in Bach [1].

relationship, we can define tractable unsupervised training loss, which we term *Maximum Distribution Kernel Entropy* (MDKE), with respect to the parameters of the encoder $f_\theta$:

$$\mathcal{L}_{\text{MDKE}}(\theta) \coloneqq -\mathcal{S}_2(\Sigma_\mathcal{D}) = \log \operatorname{tr}\left[\left(\frac{1}{M}K_\mathcal{D}\right)^2\right] = \log \sum_{i=1}^{M} \lambda_i^2\left(\frac{1}{M}K_\mathcal{D}\right) = \log \|\frac{1}{M}K_\mathcal{D}\|_F^2 \quad (14)$$

where the latter relies on the fact that the Frobenius norm $\|A\|_F^2 = \sum_i \lambda_i^2(A)$, where $\lambda_i(A)$ are eigenvalues $A$.

The MDKE objective is differentiable w.r.t. $\theta$ for commonly used kernels, provided that the encoder $f_\theta$ is differentiable as well. While the entropy estimator $\mathcal{S}_2(\Sigma_\mathcal{D})$ is convex in the kernel matrix $K_\mathcal{D}$, the objective as a whole is generally not convex in $\theta$. However, in practice, as we show in Sec. 5, mini-batch Stochastic Gradient Descent (SGD) proves to be an effective method for optimizing this objective. The effectiveness of this optimization process is significantly influenced by the parameters of the Gaussian kernels. We elaborate on the methodologies for kernel bandwidth selection in Appendix B.1.

The Frobenius norm formulation in the loss Eq. (14) significantly reduces computational complexity. However, as we have observed in some of our experiments, it can lead to the collapse of small eigenvalues of $K_\mathcal{D}$, particularly near the optimal value of the objective. To address this challenge we introduced a regularized version of the loss $\mathcal{L}_{\text{MDKE-R}}$ that incorporates optional regularization, based on the determinant $K_\mathcal{D}$ inspired by the connection with Fekete points (see details in Appendix B.2).

### 3.4 Geometrical Interpretation

The optimization objective is specifically designed to minimize the variance within each distribution (inner-distribution variance) while simultaneously maximizing the spread of distributions over the compact latent space $\mathcal{Z} = \mathcal{S}^{d-1}$. This shaping of the distributions embeddings in the latent space facilitates easier separation in downstream tasks. In this section we show that the geometry of the optimal (w.r.t. the MDKE loss) configuration of mean embeddings in the RKHS attains describe properties. For doing so we leverage the notion of *distributional variance* $\mathbb{V}_\mathcal{H}$ (Definition 1 in Muandet et al. [40]).

**Definition 3.2.** For a set of $M$ probability distributions $\mathcal{D}_M$, *distributional variance* $\mathbb{V}_\mathcal{H}(\mathcal{D}_M)$ of the mean embeddings in the RKHS $\mathcal{H}$ is given by

$$\mathbb{V}_\mathcal{H}(\mathcal{D}_M) \coloneqq \frac{1}{M}\operatorname{tr}[G] - \frac{1}{M^2}\sum_{i=1}^{M}\sum_{j=1}^{M} G_{ij}, \quad (15)$$

where $G$ is the $M \times M$ Gram matrix of mean embeddings in $\mathcal{H}$, i.e. $G_{ij} = \langle \mu_{P_i}, \mu_{P_j}\rangle_\mathcal{H}$ [40].

Here we show that the distributional variance $\mathbb{V}_\mathcal{H}$ can be equally reformulated into two separate contributions:

$$\mathbb{V}_\mathcal{H}(\mathcal{D}_M) \equiv \frac{1}{M}\sum_{i=1}^{M}\|\mu_{P_i}\|_\mathcal{H}^2 - \|\mu_{\bar{P}}\|_\mathcal{H}^2, \quad (16)$$

where $\bar{P}$ denotes mixture distribution with elements of $\mathcal{D}_M$ being uniformly weighted mixture components (see proof in the Appendix A.1).

The relevance of distributional variance for MDKE objective is established by the following result.

**Proposition 3.3.** *For a set of $M$ probability distributions $\mathcal{D}_M$, the second-order Rényi entropy $\mathcal{S}_2$ of the empirical covariance operator embedding $\hat{\Sigma}_\mathcal{D}$ induced by the choice of Gaussian distribution kernel $K_{RBF}$ over points in the RKHS $\mathcal{H}_{emb}$, - as defined in Eq. (8), - is upper bounded by the distributional variance $\mathbb{V}_{\mathcal{H}_{emb}}(\mathcal{D}_M)$, i.e.,*

$$\frac{1}{2\gamma}\mathcal{S}_2(\hat{\Sigma}_\mathcal{D}) \leq \mathbb{V}_{\mathcal{H}_{emb}}(\mathcal{D}_M) \quad (17)$$

*where $\gamma$ is the bandwidth of the distribution kernel $K_{RBF}$.*

The proof of this proposition is provided in Appendix A.3. This result formalizes the fact that our objective increases distributional variance, pushing up the average squared norm of mean embedding

of input distributions while minimizing squared norm of the mean embedding of the mixture. We further explore the geometrical implications of such optimization by formalizing connection between the variance of the distribution and the squared norm of the its mean embedding in RKHS.

**Proposition 3.4.** *Under Assumption 3.1, the maximum norm of kernel mean embedding is attained by Dirac distributions $\{\delta_z\}_{z \in \mathcal{Z}}$.*

This result is trivial due to the fact that the set of mean embeddings is contained in the convex hull of $\{k_{\text{emb}}(z, \cdot)\}_{z \in \mathcal{Z}}$, and, under Assumption 3.1, $\forall z \in \mathcal{Z} : \|k_{\text{emb}}(z, \cdot)\|^2_{\mathcal{H}_{\text{emb}}} = k_{\text{emb}}(z, z) = 1$.

**Proposition 3.5.** *Under Assumption 3.1, uniform distribution $\mathcal{U}(\mathcal{Z})$ is a unique solution of*

$$\arg\min_{P \in \mathcal{P}(\mathcal{Z})} \left\{ \|\mu_{emb}(P)\|^2_{\mathcal{H}_{emb}} \equiv \iint_{\mathcal{Z} \times \mathcal{Z}} k_{emb}(z, z') \, dP(z) \, dP(z') \right\}. \tag{18}$$

The key intuition here comes from the fact that minimization of the squared norm of mean embedding in RKHS could be seen as minimization of total interaction energy over the given surface where the potential is defined by the kernel $k$. Thus Proposition 3.5 is a special case of Theorem 6.2.1 of Borodachov et al. [7]. Similar setup was used w.r.t. Gaussian potential over the unit hypersphere in Proposition 1 from Wang and Isola [61]. The reformulation in Eq. (16) together with Propositions 3.4 and 3.5 immediately suggests that the framework could be seen as an extension of the *Hyperspherical Uniformity Gap* [32] to infinite-dimensional spaces of probability distributions This extension maintains the goal of reducing variance among input distributions while maximizing the separability between their means. See Appendix C.2 for a broader explanation of the connection.

More generally, utilizing the fact that under Assumption 3.1, the *kernel metric $d_{k_{\text{emb}}}$* (see Eq. (3)) is a monotonically increasing function of Euclidean metric on the latent space, we establish a precise connection between the generalized variance [60] and the norm of the mean embedding. Further details can be found in Appendix A.4.

An attempt to directly optimize $\mathbb{V}_{\mathcal{H}_{\text{emb}}}$ using SGD resulted in significantly weaker outcomes. While a thorough mathematical explanation necessitates further investigation, we contend that this issue aligns with the recurring challenge reported across various studies regarding direct optimization over Maximum Mean Discrepancy (MMD). We hypothesize that optimal solutions exhibit similar geometric configurations, while the entropy of the covariance operator providing a smoother objective. Nonetheless, $\mathbb{V}_{\mathcal{H}_{\text{emb}}}$ retains its value as an insightful and intuitive measure for describing the geometric configuration of the learned system.

### 3.5 An Illustrative Example

We use a simple example to illustrate the connection between geometrical configurations of embedded distributions and distribution kernel entropy $\mathcal{S}_2(\Sigma_{\mathcal{D}})$ (see Fig. 2). We sample a number of points from 6 different Gaussian distributions and project on a sphere $\mathbb{S}^2$ varying their projected variance $\gamma$. As $\gamma$ decreases, the distributional variance of the overall distribution of Gaussians increases (Fig. 2a). For very small $\gamma$ each distribution converges to a point (a Dirac distribution). This results in the entropy interpolating between lower and upper bounds, demonstrating how entropy behaves in response to changes in distribution variance. Fig. 2b showcases the behavior of two terms comprising distributional variance (Eq. (16)): the average kernel norm of the distributions alongside the kernel norm of the mixture. The increase in entropy and variance corresponds to a 'flattening' effect on the spectrum of the distribution kernel matrix. This example provides a simplified picture of how input distributions configurations influence kernel entropy.

### 3.6 Limitations

**Runtime complexity.** The applicability of a data-dependent distribution kernel to solving discriminative tasks relies on the structure of the dataset being well-suited for distribution regression modeling. The model performs best when the number of input distributions is relatively small (e.g., thousands rather than millions), while the number of samples per distribution is large. It is crucial to note that the computational complexity of the proposed method, which is a common concern in practical applications, is most favorable for the tasks described. A detailed analysis of runtime complexity can be found in Appendix B.3.

**Broader impact.** We wish to emphasize that the distribution regression framework has emerged as a powerful tool for analysis and predictive modeling, especially in domains where traditional methods face challenges, including social science, economics, and medical studies. We urge researchers and practitioners applying distribution regression in these areas to give special consideration to issues such as bias, fairness, data quality, and interpretability, - aspects that are currently under-researched in the context of distributional regression, largely due to the relative novelty.

## 4 Related Work

### 4.1 Distribution Regression

Distribution regression was introduced in Poczos et al. [45], while the seminal work of Szabó et al. [54] provides a comprehensive theoretical analysis of this regression technique. A natural approach to solving distribution regression problems involves using kernels between measures. Notable examples include the Fisher kernel [17], Bhattacharyya kernel [18], Probability product kernel [19], kernels based on nonparametric divergence estimates [52], and Sliced Wasserstein kernels [23, 37]. Muandet et al. [39] proposed leveraging the mean embedding of measures in RKHS, and Szabó et al. [53] provided theoretical guarantees for learning a ridge regressor from distribution embeddings in Hilbert space to the outputs. Distribution kernels have been successfully applied in various kernel-based methods, such as SVM [39], Ridge Regression [37], Bayesian Regression [28], and Gaussian Process Regression [2]. They have also been adapted for different modalities like distribution to distribution regression [44], sequential data [29], and more. Some learning paradigms can be considered closely related to distributional classification settings, such as multiple instance learning, where group-level information (i.e., labels) is available during training [70, 26, 25]. For an in-depth exploration of the diverse methodologies employed in distributional regression settings, we invite readers to consult Appendix C.1.

### 4.2 Matrix Information Theory

Quantum entropy, including Rényi entropy, is a powerful metric to describe information in a unique way (see Müller-Lennert et al. [42] for foundational insights). Giraldo et al. [15] designed the measure of entropy using operators in RKHS to mimic Rényi entropy's behavior, offering the advantage of direct estimation from data. Bach [1] applied von Neumann entropy of the density matrix to the covariance operator embedding of probability distributions, thereby defining an information-theoretic framework utilizing kernel methods. In machine learning, especially within self-supervised learning (SSL) setups, entropy concepts have recently found novel applications. Our study builds on most recent developments [49, 21, 55] by applying quantum Rényi entropy to the covariance operator in RKHS.

## 5 Experiments

We here demonstrate that our proposed method successfully performs unsupervised learning of data-dependent distribution kernel across different modalities. The experimental setup is divided into two phases: unsupervised pre-training and downstream regression classification using the learned kernel.

For each dataset, we select a hold-out validation subset with balanced classes, while the remainder of the dataset is utilized for unsupervised pre-training. We use mini-batch ADAM [22] with a static learning rate of $0.0005$. We report mini-batch based (instead of epoch based) training dynamics as our tasks do not require cycling over the entire dataset to converge to the optimal loss value. All experiments use Gaussian kernel both as an *embedding kernel* and *distribution kernel*, the hyperparameter selection is performed as described in Appendix B.1.

Once the samples encoder $f_\theta$ is learned, we employ it to compute distribution kernel Gram matrix, used as an input to the Support Vector Machine (SVM) for solving downstream classification tasks. A grid search with 5 splits (70/30) is conducted to optimize the strength of the squared $l_2$ regularization penalty $C$, exploring 50 values over the log-spaced range $\{10^{-7}, \ldots, 10^5\}$. The best estimator is then applied to evaluate classification accuracy on the validation subset, which we report.

Additional experiments exploring the application of data-dependent distribution kernels in domains where distribution regression models are less common, such as image and text, are presented in Appendix D.

## 5.1 Flow Cytometry

Flow Cytometry (FC) is a widely used technique for measuring chemical characteristics of mixed cell population. Because population-level properties are described through (randomized) sampling of cells, FC is used as a canonical setup of distribution regression. For this study we used a dataset [56] where more than $100.000$ cells are measured per each patient (subject). For each cell a total of ten parameters are reported, hence, we treated each subject as an empirical distribution over $\mathbb{R}^{10}$. We considered downstream classification tasks on two different sets of labels. The first ('Tissue' classification) contains peripheral blood (pB) and bone marrow (BM) samples from $N = 44$ subjects. The second ('Leukemia' classification) presents healthy and leukemia BM cell samples, $N = 50$. Classes were balanced in both cases. We sampled 16 subjects for Tissue classification and 20 subjects for Leukemia for training. Unsupervised learning was performed over the entire dataset. The encoder $f_\theta$ was parametrized by a 2-layers neural network (NN) with ReLU nonlinearity and $l_2$ normalized output (on the unit hypersphere $\mathbb{S}^9$). Per each subject we sampled a small percentage of cells, and we report performance for the sample size of 200. We repeated each training and testing phase for 100 times to track the variance induced by this aggressive subsampling.

Table 1: Distribution regression accuracy on Flow Cytometry datasets.

| MODEL | TISSUE | | LEUKEMIA | |
|---|---|---|---|---|
| | ACC. | VAR. | ACC. | VAR. |
| GMM-FV | 93.07% | ±0.308 | 94.80% | ±0.186 |
| SW1 | 87.10% | ±0.530 | 95.07% | ±0.111 |
| SW2 | 81.71% | ±0.341 | **95.30%** | ±0.224 |
| MMD LINEAR | 82.42% | ±0.840 | 90.57% | ±0.208 |
| MMD GAUSSIAN | 81.71% | ±0.574 | 92.23% | ±0.216 |
| MMD CAUCHY | 81.57% | ±0.662 | 93.77% | ±0.080 |
| MMD IMQ | 82.89% | ±0.698 | 91.43% | ±0.217 |
| DISTR. VAR. | 79.47% | ±0.011 | 91.82% | ±0.007 |
| MDKE RAND | 77.50% | ±0.002 | 89.50% | ±0.003 |
| MDKE NO REG. | 95.30% | ±0.002 | 92.46% | ±0.002 |
| MDKE REG. | **98.89%** | ±0.010 | 94.57% | ±0.005 |

To demonstrate the impact of unsupervised pre-training, we compared several methods across multiple configurations (Table 1):

a) **Kernels on distributions.** This group includes Fisher kernels applied to parametric Gaussian Mixture Model (GMM) estimates, as suggested in Krapac et al. [24], along with Sliced Wasserstein-1 and Sliced Wasserstein-2 kernels [37].

b) **MMD kernels.** Here, we employ a Gaussian embedding kernel for mean embeddings, marked as 'MMD' (Maximum Mean Discrepancy) in the table. This category includes various options for the distribution kernel, such as linear, Gaussian, Cauchy, and inverse multiquadrics.

c) **Distributional Variance.** An ablation study is conducted to demonstrate the results of directly optimizing the *distributional variance* defined in Eq. (15).

d) **MDKE.** We explore various configurations of the encoder optimized with the MDKE objective. We report performance for randomly initialized encoder, and for unsupervised pre-trained encoder with and without regularization. Random initialization happens only once, and all subsequent accuracy measurements are taken using the same encoder.

The variance reported for each model is measured across multiple runs to demonstrate the effect of the sampling. Importantly, the optimization of the MDKE objective results in embedding kernels with significantly lower variance compared to non-data-specific kernels.

## 5.2 Image and Text Modalities

In the following section, we present additional experiments on learning data-dependent distribution kernels in domains not typically considered distributional regression tasks, specifically the image and text domains. While we acknowledge the existence of more powerful domain-specific models and methods for both modalities, we provide these results to demonstrate the framework's applicability across a wide range of settings under appropriate choice of the representation model. Representing text as empirical samples from the finite space of tokens (i.e. words from the dictionary) is quite

common, while the choice to model images as histograms over pixel positions is more subtle. We demonstrate that, in both scenarios, unsupervised pre-training of the encoder yields distribution kernel that achieves strong performance on downstream classification tasks, showcasing the versatility of the proposed learning framework in scenarios where distribution regression formulations are uncommon.

**Images.** MNIST [12] and Fashion-MNIST [65] consist of $28 \times 28$ pixel grayscale images divided into 10 classes. We considered each individual image to be a probability distribution (via rescaling pixel intensities so that $l_1(\text{image}) = 1$) over the discrete space of pixel positions (i.e., histogram). Given this support space, the encoder $f_\theta$ is a discrete map which we implemented as a table lookup (i.e., embeddings) from pixel indices to the points on the hypersphere $\mathcal{S}^{31}$. Embeddings were initialized by sampling points uniformly. Gradients of the MDKE objective with respect to the embeddings parameters were computed via automatic differentiation using projected gradient steps to ensure that the embeddings remain on the hypersphere. The small size of the support space enables the *exact* computation of the inner product between kernel mean embeddings of input distributions Eq. (7) and, subsequently, the distribution kernel Eq. (8) during both training and evaluation. This ensured a lower variance of the accuracy for the downstream classification. Performing MNIST classification upon pre-training with our unsupervised encoder significantly improves the baseline (random initialization of latent embeddings) accuracy of $85.0\%$ by reaching a plateau at $92.15\%$. Refer to the detailed analysis of the latent spaces of the trained encoders in Appendix D.1.

**Text.** To assess our method's performance in a larger discrete support space, we utilized the "20 Newsgroups" [27], a multi-class text classification dataset. We reduced the size of the dataset to 5 classes (resulting in $2,628$ sentences and $38,969$ unique words) by subsampling both training and test subsets. We treated sentences as empirical distributions over words, assuming word sets to be enough for topic classification, despite no positional info. The encoder $f_\theta$ mirrored the setup used in the MNIST case (Appendix D.1), with $l_2$ normalized word embeddings on $\mathbb{S}^{31}$. However, while in the MNIST case embeddings computations were performed *exactly*, here considering the entire *embedding kernel* Gram matrix is impractical due to its large size. Instead, we optimized embeddings by randomly sampling 20 words per sentence, making the inner product between embeddings a *stochastic approximation*. This setup is meant to confirm that the optimization of the proposed MDKE objective yields a solution that is robust w.r.t. the excessive risk induced by first-level subsampling. Detailed results can be found in Appendix D.2.

## 6 Conclusion

In this work, we presented an unsupervised way of learning data-dependent distribution kernel. While previous studies in distribution regression predominantly relied on hand-crafted kernels, our work, in contrast, demonstrates that entropy maximization can serve as a powerful guiding principle for learning adaptable, data-dependent kernel in the space of distributions. Our empirical findings show that this technique can not only serve as a pre-training step to enhance the performance of downstream distribution regression tasks, but also facilitate complex analyses of the input space. The interpretation of the learning dynamics induced by the proposed objective relies on a theoretical link between the quantum entropy of the dataset embedding and distributional variance. This theoretical link, which we have proven, enables us to approach the optimization from a geometrical perspective, providing crucial insights into the flexibility of the learned latent space encoding.

We hope that theoretically grounded way of learning data-dependent kernel for distribution regression tasks will become a strong alternative to the common practice of hand-picking kernels. More broadly, our results present a methodology for leveraging the distributional nature of input data along side the novel perspective on the encoding of complex input spaces. This highlights the potential to extend the application of more advanced learning methods, embracing the ever-increasing complexity of data by going beyond more conventional vector-based representations.

**Acknowledgments.** We thank the Allen Institute for Brain Science founder, Paul G. Allen, for his vision, encouragement, and support.

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

# A Proofs

In this section, we present the proofs for the propositions outlined in our study. To ensure clarity, we will first restate the setup and introduce necessary concepts.

We define the input space as $\mathcal{X}$, and $\mathcal{P}(\mathcal{X})$ represents the space of probability distributions over $\mathcal{X}$. Consider a dataset of $M$ probability distributions, denoted as $\mathcal{D}_M = \{P_i \in \mathcal{P}(\mathcal{X})\}_{i=1}^M$. With a p.d. characteristic *embedding kernel* $k : \mathcal{X} \times \mathcal{X} \to \mathbb{R}$, the corresponding RKHS $\mathcal{H}$, and the feature map $\phi : \mathcal{X} \to \mathcal{H}$, we define the *mean embedding map* $\mu : \mathcal{P}(\mathcal{X}) \to \mathcal{H}$ such that $\mu_P = \mu(P) := \int_{\mathcal{X}} \phi(x) \, dP(x)$. A p.d. translation-invariant characteristic *distribution kernel* $K : \mathcal{P}(\mathcal{X}) \times \mathcal{P}(\mathcal{X}) \to \mathbb{R}$ is defined using the mean embeddings of corresponding distributions. For simplicity, $\mu_i$ denotes $\mu(P_i)$.

Additional concepts essential for our proofs include the Gram matrix of mean embeddings $G \in \mathbb{R}^{M \times M}$, representing the inner products of mean embeddings in the dataset, i.e., $G := [\langle \mu_i, \mu_j \rangle_{\mathcal{H}}]_{ij}$. The kernel matrix $K_{\mathcal{D}} \in \mathbb{R}^{M \times M}$ with respect to the *distribution kernel* $K$ is denoted as $K_{\mathcal{D}} := [K(P_i, P_j)]_{ij}$. We also recall the definition of *distributional variance* $\mathbb{V}_{\mathcal{H}}$ (see Eq. (15)):

$$\mathbb{V}_{\mathcal{H}}(\mathcal{D}_M) := \frac{1}{M} \text{tr}[G] - \frac{1}{M^2} \sum_{i=1}^M \sum_{j=1}^M G_{ij}$$

These definitions and notations will be referenced throughout the proofs.

## A.1 Kernel Norms Gap and Distributional Variance

For both cases of input distributions being empirical probability distributions or continuous densities, we define mixture distribution $\bar{P}$, with a slight abuse of notation:

$$\bar{P}(x) := \frac{1}{M} \sum_{i=1}^M P_i(x)$$

**Lemma A.1.** *For the mixture distribution $\bar{P}$ and the Gram matrix $G$, the following relationship holds:*

$$\|\mu_{\bar{P}}\|_{\mathcal{H}}^2 = \frac{1}{M^2} \sum_{i=1}^M \sum_{j=1}^M G_{ij} \tag{19}$$

*Proof.* We begin by recalling that the inner product between mean embeddings $\mu_i$ and $\mu_j$ in $\mathcal{H}$ is given by:

$$\langle \mu_i, \mu_j \rangle_{\mathcal{H}} = \iint_{\mathcal{X} \times \mathcal{X}} k(x, x') \, dP_i(x) \, dP_j(x')$$

Substituting this into the expression for the Gram matrix, we have:

$$\frac{1}{M^2} \sum_{i=1}^M \sum_{j=1}^M G_{ij} = \frac{1}{M^2} \sum_{i=1}^M \sum_{j=1}^M \iint_{\mathcal{X} \times \mathcal{X}} k(x, x') \, dP_i(x) \, dP_j(x')$$

$$= \iint_{\mathcal{X} \times \mathcal{X}} k(x, x') \, d\left(\frac{1}{M} \sum_{i=1}^M P_i(x)\right) d\left(\frac{1}{M} \sum_{j=1}^M P_j(x')\right)$$

$$= \iint_{\mathcal{X} \times \mathcal{X}} k(x, x') \, d\bar{P}(x) \, d\bar{P}(x') = \|\mu_{\bar{P}}\|_{\mathcal{H}}^2$$

This completes the proof. $\qquad\square$

By incorporating Eq. (19) into the definition of distributional variance (see Eq. (15)), and noting that the trace of $G$ is the sum of squared norms of input distributions (i.e., $\text{tr}[G] = \sum_{i=1}^M \|\mu_{P_i}\|_{\mathcal{H}}^2$), we obtain:

$$\mathbb{V}_{\mathcal{H}}(\mathcal{D}_M) \equiv \frac{1}{M} \sum_{i=1}^M \|\mu_{P_i}\|_{\mathcal{H}}^2 - \|\mu_{\bar{P}}\|_{\mathcal{H}}^2$$

This result offers a new and intuitive perspective on distributional variance, conceptualizing it as the difference between the average squared norm of individual input distributions and the squared norm of the mixture distribution.

## A.2 Pairwise Distance and Distributional Variance

**Definition A.2.** For a dataset $\mathcal{D}_M$ of $M$ probability distributions, we define the average pairwise distance between kernel mean embeddings in $\mathcal{H}$ as $\mathbb{J}_{\mathcal{H}}(\mathcal{D}_M)$, given by:

$$\mathbb{J}_{\mathcal{H}}(\mathcal{D}_M) := \frac{1}{M^2} \sum_{i=1}^{M} \sum_{j=1}^{M} ||\mu_i - \mu_j||_{\mathcal{H}}^2 \tag{20}$$

**Lemma A.3.** *For the distributional variance $\mathbb{V}_{\mathcal{H}}$ of the dataset $\mathcal{D}_M$, the following relationship holds:*

$$\mathbb{V}_{\mathcal{H}}(\mathcal{D}_M) \equiv \frac{1}{2} \cdot \mathbb{J}_{\mathcal{H}}(\mathcal{D}_M) \tag{21}$$

*Proof.* Starting with the definition of $\mathbb{J}_{\mathcal{H}}(\mathcal{D}_M)$:

$$\mathbb{J}_{\mathcal{H}}(\mathcal{D}_M) = \frac{1}{M^2} \sum_{i=1}^{M} \sum_{j=1}^{M} ||\mu_i - \mu_j||_{\mathcal{H}}^2$$

$$= \frac{1}{M^2} \sum_{i=1}^{M} \sum_{j=1}^{M} \langle \mu_i - \mu_j, \mu_i - \mu_j \rangle_{\mathcal{H}}$$

$$= \frac{1}{M^2} \sum_{i=1}^{M} \sum_{j=1}^{M} \left( \langle \mu_i, \mu_i \rangle_{\mathcal{H}} + \langle \mu_j, \mu_j \rangle_{\mathcal{H}} - 2 \langle \mu_i, \mu_j \rangle_{\mathcal{H}} \right)$$

$$= \frac{1}{M^2} \left( 2M \sum_{i=1}^{M} \langle \mu_i, \mu_i \rangle_{\mathcal{H}} - 2 \sum_{i=1}^{M} \sum_{j=1}^{M} \langle \mu_i, \mu_j \rangle_{\mathcal{H}} \right)$$

$$= 2 \left( \underbrace{\frac{1}{M} \sum_{i=1}^{M} \langle \mu_i, \mu_i \rangle_{\mathcal{H}}}_{\text{diagonal of } G} - \frac{1}{M^2} \sum_{i=1}^{M} \sum_{j=1}^{M} \langle \mu_i, \mu_j \rangle_{\mathcal{H}} \right)$$

Recognizing that $\sum_{i=1}^{M} \langle \mu_i, \mu_i \rangle_{\mathcal{H}} = \text{tr}[G]$, we can equate expression in the brackets to $\mathbb{V}_{\mathcal{H}}(\mathcal{D}_M)$, leading to:

$$\mathbb{V}_{\mathcal{H}}(\mathcal{D}_M) \equiv \frac{1}{2} \cdot \mathbb{J}_{\mathcal{H}}(\mathcal{D}_M),$$

which concludes the proof. $\square$

Lemma A.3 will be instrumental in further proofs, establishing a crucial link between distributional variance in $\mathcal{H}$ and *quantum entropy* of the covariance operator embedding $\Sigma_{\mathcal{D}}$.

## A.3 Distribution Kernel Entropy Upper-Bound

In this section, we provide the proof for the key theoretical result stated in Proposition 3.3.

Consider a dataset $\mathcal{D}$ consisting of probability distributions $\{P \in \mathcal{P}(\mathcal{X})\}_i$ sampled i.i.d. from an unknown meta-distribution $\mathbb{D}$. We assert that the second-order Rényi entropy $\mathcal{S}_2$ of the empirical covariance operator embedding $\Sigma_{\mathcal{D}}$, induced by the choice of Gaussian *distribution kernel* $K_{\text{RBF}}$ over points in the RKHS $\mathcal{H}$, is upper-bounded by the distributional variance $\mathbb{V}_{\mathcal{H}}(\mathcal{D})$:

$$\frac{1}{2\gamma} \mathcal{S}_2(\hat{\Sigma}_{\mathcal{D}}) \leq \mathbb{V}_{\mathcal{H}}(\mathcal{D})$$

where $\gamma$ is the bandwidth of the *distribution kernel* $K_{\mathrm{RBF}}$.

Starting from the properties of $\mathcal{S}_2(\hat{\Sigma}_{\mathcal{D}})$ stated in Eq. (13):

$$\mathcal{S}_2(\hat{\Sigma}_{\mathcal{D}}) = \mathcal{S}_2\left(\frac{1}{M}K_{\mathcal{D}}\right) = -\log\sum_{i=1}^{M}\sum_{j=1}^{M}\left(\frac{1}{M}K_{ij}\right)^2 = -\log\left(\frac{1}{M^2}\sum_{i=1}^{M}\sum_{j=1}^{M}K_{ij}^2\right) \qquad (22)$$

Applying Jensen's inequality, considering the concavity of the $\log$, to Eq. (22), we have:

$$\mathcal{S}_2(\hat{\Sigma}_{\mathcal{D}}) \leq -\frac{1}{M^2}\sum_{i=1}^{M}\sum_{j=1}^{M}\log K_{ij}^2$$

$$= -\frac{1}{M^2}\sum_{i=1}^{M}\sum_{j=1}^{M}\log\exp\left(-\frac{\gamma}{2}\|\mu_i - \mu_j\|_{\mathcal{H}}^2\right)^2 \qquad (23)$$

$$= -\frac{1}{M^2}\sum_{i=1}^{M}\sum_{j=1}^{M}\log\exp\left(-\gamma\|\mu_i - \mu_j\|_{\mathcal{H}}^2\right)$$

$$= \gamma\left(\frac{1}{M^2}\sum_{i=1}^{M}\sum_{j=1}^{M}\|\mu_i - \mu_j\|_{\mathcal{H}}^2\right)$$

$$= \gamma \cdot \mathbb{J}_{\mathcal{H}}(\mathcal{D}) \qquad (24)$$

$$= 2\gamma \cdot \mathbb{V}_{\mathcal{H}}(\mathcal{D}) \qquad (25)$$

Eq. (23) uses the definition of $K_{\mathrm{RBF}}$ from Eq. (8), Eq. (24) is derived from the definition of the average pairwise distance in Eq. (20), and Eq. (25) follows from Lemma A.3. This completes the proof of Proposition 3.3, establishing the upper bound for the second-order quantum Rényi entropy of the covariance operator embedding $\Sigma_{\mathcal{D}}$ in terms of the distributional variance in RKHS $\mathcal{H}$. $\qquad\square$

*Remark A.4.* It is important to note that $\mathcal{S}_2$ is typically measured using the base-2 logarithm ($\log_2$) rather than the natural logarithm. However, the proof remains accurate with a proper re-scaling to account for the change in the logarithmic base.

## A.4 Generalized Variance in RKHS

In this section we prove the connection between generalized variance and norm of the kernel mean embedding.

So far we established the squared norm of mean embedding is maximized by Dirac points (zero variance) and is minimized by uniform distribution (max variance). We now formalize the intuition that larger squared norm in embedding kernel RHKS corresponds to smaller variance in the latent space. We first note that under Assumption 3.1, the *kernel metric* $d_{k_{\mathrm{emb}}}$ (see Eq. (3)) induced by the choice of $k_{\mathrm{emb}}$ is a monotonically increasing function of Euclidean metric on the latent space, thus is representative of the geometry of encoded input distributions. Which let us use a generalized notion of variance here.

**Definition A.5.** (Vakhania et al. [60]) Let $X$ be a random variable which takes values in a Fréchet space $\mathcal{F}$ equipped with seminorm $\|\cdot\|_{\alpha}$. And suppose that $X$ is square-integratable, in a sense that $\mathbb{E}\|X\|_{\alpha}^2 < \infty$. Let $\mu \in \mathcal{F}$ be a Pettis integral of $X$ (i.e. generalization of the mean). Generalized variance of $X$ w.r.t. seminorm $\|\cdot\|_{\alpha}$ is defined as following

$$\mathrm{Var}_{\alpha}[X] := \mathbb{E}\|X - \mu\|_{\alpha}^2. \qquad (26)$$

Note that for a random variable in RKHS defined as a push-forward of a probability distribution $P \in \mathcal{P}(\mathcal{Z})$ over the latent space, i.e. $X = z \sim \phi\#P$ satisfies conditions of Definition A.5 with $\mathcal{F} = \mathcal{H}$, $\|\cdot\|_{\alpha} = \|\cdot\|_{\mathcal{H}_{\mathrm{emb}}}$, and a Pettis integral being a kernel mean embedding. Denote the described variance as $\mathrm{Var}_{\mathcal{H}_{\mathrm{emb}}}[P]$. We now show that

**Proposition A.6.** *Under Assumption 3.1, for every $P \in \mathcal{P}(\mathcal{Z})$,*

$$Var_{\mathcal{H}_{emb}}[P] = 1 - \|\mu_{emb}(P)\|_{\mathcal{H}_{emb}}^2. \qquad (27)$$

*Proof.* From the definition of generalized variance we have the following

$$
\begin{aligned}
\mathrm{Var}_{\mathcal{H}_{\mathrm{emb}}}[P] &= \mathbb{E}_{x \sim P} \| \phi(x) - \mu_P \|_{\mathcal{H}_{\mathrm{emb}}}^2 \\
&= \int \| \phi(x) - \mu_P \|_{\mathcal{H}_{\mathrm{emb}}}^2 \, dP(x) \\
&= \int \langle \phi(x) - \mu_P, \phi(x) - \mu_P \rangle_{\mathcal{H}_{\mathrm{emb}}} \, dP(x) \\
&= \int \left( \| \phi(x) \|_{\mathcal{H}_{\mathrm{emb}}}^2 + \| \mu_P \|_{\mathcal{H}_{\mathrm{emb}}}^2 - 2 \langle \phi(x), \mu_P \rangle_{\mathcal{H}_{\mathrm{emb}}} \right) \, dP(x) \\
&= \int \| \phi(x) \|_{\mathcal{H}_{\mathrm{emb}}}^2 \, dP(x) + \int \| \mu_P \|_{\mathcal{H}_{\mathrm{emb}}}^2 \, dP(x) - 2 \int \langle \phi(x), \mu_P \rangle_{\mathcal{H}_{\mathrm{emb}}} \, dP(x) \\
&= \int 1 \, dP(x) + \| \mu_P \|_{\mathcal{H}_{\mathrm{emb}}}^2 \int 1 \, dP(x) - 2 \left\langle \int \phi(x) \, dP(x), \mu_P \right\rangle_{\mathcal{H}_{\mathrm{emb}}} \\
&= 1 + \| \mu_P \|_{\mathcal{H}_{\mathrm{emb}}}^2 - 2 \| \mu_P \|_{\mathcal{H}_{\mathrm{emb}}}^2 \\
&= 1 - \| \mu_P \|_{\mathcal{H}_{\mathrm{emb}}}^2,
\end{aligned}
\tag{28}
$$

which concludes the proof. $\qquad\square$

The proposition demonstrates the motivation behind training the encoder to minimize the variance in the latent space by maximizing the (average) squared norm of the mean embeddings.

## B  Practical Aspects of Learning

### B.1  Kernel Hyperparameter Selection

In practice, the effectiveness of the optimization process is significantly influenced by the parameters of the Gaussian kernels. Setting the bandwidth parameter $\gamma$ either too low or too high can hinder the model's ability to learn effectively. This issue is a well-known challenge in working with kernel methods, where determining the optimal kernel bandwidth has been an area of extensive study.

In our work, we have employed an empirical approach, which involves adjusting $\gamma$ based on the idealized structure of the dataset after its projection onto the latent space. Specifically, we set $\gamma$ such that $1/\gamma$ equals 10 times the average distance to the nearest neighbor in the set of points sampled uniformly on $\mathbb{S}^{d-1}$ (inspired by the experimental approach in Blanchard et al. [4]). The number of points is chosen to be the number of distributions in the training set for the distribution kernel. For the embedding kernel, it is set to the number of distributions in a batch multiplied by the number of samples per distribution used for unsupervised training.

### B.2  Optional Regularization

As the theoretical maximum of $\mathcal{S}_2$ entropy is attained when the spectrum of $\frac{1}{M} K_{\mathcal{D}}$ is uniform, the optimal encoder with respect to $\mathcal{L}_{\mathrm{MDKE}}$ also tends to maximize the determinant of the kernel Gram matrix $K_{\mathcal{D}}$. This relationship is intuitive when considering that the determinant of a matrix is the product of its eigenvalues. The points that maximize such determinant $K_{\mathcal{D}}$ are theoretically known as *Fekete points* [35, 3] and, in our case, are relative to the *distribution kernel*. Fekete points have been shown by Karvonen et al. [20] to be an optimal configuration for learning kernel interpolants, making them particularly suitable for downstream tasks framed as kernel regression. As such, encoders optimized under the $\mathcal{L}_{\mathrm{MDKE}}$ objective facilitate more accurate and robust performance in subsequent regression tasks.

In practice, we found that optimizing the MDKE objective posed certain numerical challenges, particularly due to the tendency of too small eigenvalues in the distribution kernel matrix $K_{\mathcal{D}}$ to collapse near the optimal value of the objective when using large batch size for training. To mitigate this issue and prevent undesirable optimization behavior, we have introduced a regularization term to the original objective. This term is inspired by the concept of Fekete points configuration, leading to

the following loss function:

$$\mathcal{L}_{\text{MDKE-R}}(\theta) := -\mathcal{S}_2(\Sigma_{\mathcal{D}}) + \epsilon \cdot \Omega(\Sigma_{\mathcal{D}})$$

$$= \log \| \frac{1}{M} K_{\mathcal{D}} \|_F^2 - \epsilon \cdot \log \det \left| \frac{1}{M} K_{\mathcal{D}} \right| \tag{29}$$

Here, $\epsilon$ serves as a hyperparameter to control the strength of regularization. The regularization term $\Omega(\Sigma_{\mathcal{D}})$ is designed to stabilize the optimization process by counteracting the effects of the collapse of small eigenvalues. The empirical evidence supporting the effectiveness of regularization in stabilizing the training dynamics for the sentence representation learning experiment (as detailed in Appendix D.1) is showcased in Fig. 3.

### B.3 Runtime Complexity

Scalability of the kernel methods is typically a key concern when it comes to practical applications. The runtime complexity of the proposed method could be decomposed into two components:

- Computation of the distribution kernel Gram matrix. This involves computing $O(N^2)$ inner products between distributions where $N$ is a number of distributions. Each inner product involves computing matrix of pairwise distances between samples from each distribution, in the case of computing the Gaussian kernel over points on the hypersphere, the runtime complexity is the complexity of multiplying matrices $\mathbb{R}^{M \times d}$ where $M$ is a number of samples. All computations could be efficiently parallelized.

- Solving regression using distribution kernel Gram matrix. The complexity comes from matrix inverse and is typically estimated to be $O(N^3)$ with a range of methods proposed to reduce the complexity [8, 47, 64, 36, 31].

With this being said, we want to highlight that the runtime complexity of the proposed method is most favorable exactly for those tasks and datasets where distributional regression is an appropriate modeling approach. A significant number of samples per distribution ensures a high accuracy in approximating the kernel between a pair of distributions, and, at the same time, a relatively small number of distributions in the dataset alleviates issues related to storing the distribution kernel Gram matrix in memory and performing computations on it

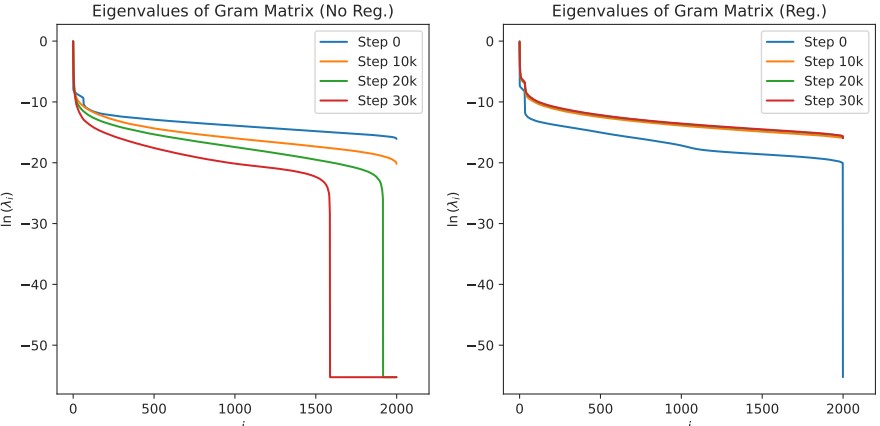

Figure 3: **The effect or regularization on the training dynamics.** The distribution of the eigenvalues of the distribution kernel Gram matrix, calculated for 2,000 sentences sampled from '20 Newsgroups' dataset (details in Appendix D.2), is observed throughout the training. (a) Training with no regularization leads to the collapse of smaller eigenvalues. (b) The regularization stabilizes the training by preventing eigenvalues from collapsing.

# C  Connection to Other Frameworks

## C.1  Distributional Regression Landscape

As the distributional regression task differs from common Machine Learning (ML) setups where inputs are given as vectors, the effective practical solution requires unique considerations.

The most obvious approach would be to ignore the fact that inputs are given as distributions and learn classifier on the space of samples, with a proper aggregation of posteriors (e.g. with a simple sum over histograms). This approach, while being simple, has been shown not to yield practically useful results, and was explicitly excluded from reported performance on different tasks by different authors [39, 46].

Viable approaches to solving distributional regression could be, approximately, split up into the following categories.

**Discriminate generative models.**   The idea is to fit each input distribution to a parametric family, e.g. Gaussian Mixture Model (GMM), to use available closed-form solutions to compute kernel or similarity or distance between distributions. This group of methods originated from the work on hidden Markov models and it's applications to processing sequence modalities, like text, DNA, proteins, and more. Early work [17] on learning discriminative classifier for generative models leveraged the fact that parametric models forms a Riemannian manifold with local metric given by Fisher information, they derived kernel function termed Fisher kernel suitable for running SVM between generative models. Driving motivation was classification between hidden Markov models, with the developed method being applied to DNA and protein sequence analysis. Following the same modeling approach of analysing DNA sequences with discriminative models between Markov models, Jebara and Kondor [18] proposed to use Bhattacharyya distance between distributions from exponential families to derived so-called Expected Likelihood Kernel. Jebara et al. [19] explored the method of computing kernel between distributions as the integral of the product of pairs of distributions, termed Probability Product Kernel. A new family of kernels was applied to the same setup of discriminating task on text modality, hidden Markov models for biological data. The model was also successfully applied for analysis of linear dynamical systems for time series data. Critical advantage of this work was access to computationally effective way of computing kernel for the distributions without having access to analytical closed-form parametrization (only relying on samples). In Sec. 5.1 we used fitting distributions to GMM with Fisher Kernel applied to learn parameters as a baseline for the performance on the task.

**Point clouds.**   This group includes methods that model each input distribution as a set of points (also known as 'point cloud' or 'feature group' in computer vision) and use kernel or similarity function defined on sets. Large portion of the methods in this group arised in computer vision (CV) field when local features extractor were widely employed to pre-process images yielding either per-image histograms or sets of low dimensional vectors. The group includes kernels based on nonparametric divergence estimates, quantized set kernels, and so-called 'nearest-neighbor' set kernels to name a few [52, 30, 34, 16, 5, 5, 59]. Such kernels were employed to many CV tasks though the successfully application required not only a good kernel but also a high-quality feature extractor. Special attention goes to methods leveraging Wasserstein distance (including both kernel-based and similarity-based solutions). Wasserstein distance as a metric-aware discrepancy measure for probability distributions is a natural choice of deriving kernels or similarity functions between point clouds [46]. While being computationally problematic for large-scale problems, Sliced Wasserstein kernels were succesfully adopted [23, 37] as they provided both reliable way for the point set comparison with a reduced cost leveraging sliced formulation. In Sec. 5.1 we compared performance of Sliced Wasserstein-1 and Sliced Wasserstein-2 kernels, with the latter yielding significantly lower performance. Such a behavior is consistent with theoretical analysis stating that Sliced Wasserstein-1 has favoriable properties when compared to kernel based on Wasserstein-2 distance.

**Kernel Mean Embeddidngs.**   Muandet et al. [39] proposed to leverage kernel mean embedding of measures in RKHS, so that the distributional regression could now be casted to a regression the corresponding Hilbert space. While the original work leveraged the kernel between the RKHS embeddings to train SVM, Szabó et al. [53] provided theoretical guarantees for the for learning Ridge regressor from distribution embeddings in the Hilbert space to the outputs. Law et al. [28] applied

the same approach in Bayesian regression settings, and [2] used it for Gaussian Process Regression. The setup gives a great deal of flexibility by choosing the kernel in RKHS, with a few being tested in practice. Gaussian kernel is a common choice, due to its universality [11]. Among others, inverse multiquadric (IMQ) is a popular choice, typically paired with random features to improve the runtime complexity of the algorithm. Linear kernel in RKHS, despite not being universal, was reported to produce competitive results in multiple practical setups. In Sec. 5.1 we compared performance for 4 different kernels in RKHS, namely linear, Gaussian, Cauchy, and IMQ, all defined as functions of Maximum Mean Discrepancy (MMD), which gives as a way to compute the value of the kernel from finite number of samples with high accuracy. While all kernels demonstrated different performance in terms of the test accuracy, the difference reported is not that substantial.

Distribution kernels were successfully applied in different kernel-based methods and in different modalities, like distribution to distribution regression [44], distribution regression for sequential data [29], and more. In cases where input data is naturally represented as a probability distribution but not immediately applicable for existing distribution regression solutions, pre-processing or encoding of inputs is required. Yoshikawa et al. [68] proposed learning a latent space representation for input data to apply distribution regression in the text modality, with the encoder being trained jointly with the classifier. To our best knowledge, methods for learning data-dependent kernel for solving distributional regressions were not previously reported.

### C.2 Hyperspherical Uniformity Gap

The link between entropy maximization in the space of distribution, the gap between average norm and norm of the mixture distribution marginalized over the dataset (i.e. 'average' distribution), and properties of distributions that minimize and maximize kernel mean embedding norm (as described in Sec. 3.4) unveils a subtle yet significant connection to the concept of the *Hyperspherical Uniformity Gap* (HUG), introduced in [32]. HUG has been developed to generalize the phenomenon of neural collapse observed in supervised classification settings. In our approach, working directly with samples from input distributions grants us explicit access to the 'grouping' of points in the input space. Even though the HUG framework setup does not have a notion of 'grouping', we can close the gap by noting that class labels provided with the dataset implicitly create 'groupings' of points, which can be interpreted as empirical samples drawn from latent probability distributions (one for each class).

However, a notable distinction lies in the dimensional aspect: HUG focuses on the distribution of points on the surface of a finite-dimensional hypersphere, whereas our work encompasses an infinite-dimensional hyperball setting, given the use of points in kernel-induced RKHSs. Furthermore, the loss function introduced in our study presents a unified optimization objective that weaves together both the inner-group and inter-group dynamics while being derived from first principles. In the HUG framework these two aspects are addressed separately.

Establishing a more formal connection between these frameworks emerges as a promising direction for future research. Such an endeavor could offer a novel perspective on supervised classification, particularly by conceptualizing *class prototypes* as probability distributions rather than mere vectors. This exploration might bridge the gap between these distinct approaches, enriching our understanding of classification paradigms in high-dimensional spaces.

## D  Additional Experiments

### D.1 Image Classification Tasks

Performing MNIST classification upon pre-training with our unsupervised encoder significantly improves the baseline (random initialization of latent embeddings) accuracy of $85.0\%$ by reaching a plateau at $92.15\%$.

To understand the data-dependency of our encoding procedure, we analyzed the latent spaces of MNIST and Fashion-MNIST datasets. The visualization of pixel-level interaction, computed using a Gaussian kernel Gram matrix, reveals complex, dataset-specific interactions (Fig. 4a, 4c). Spectral clustering using the kernel Gram matrix provides deeper insight into the pixel interaction landscape. The chart shows clusters of pixels with correlated intensities (Fig. 4b, 4d), with the number of clusters set to 10 empirically.

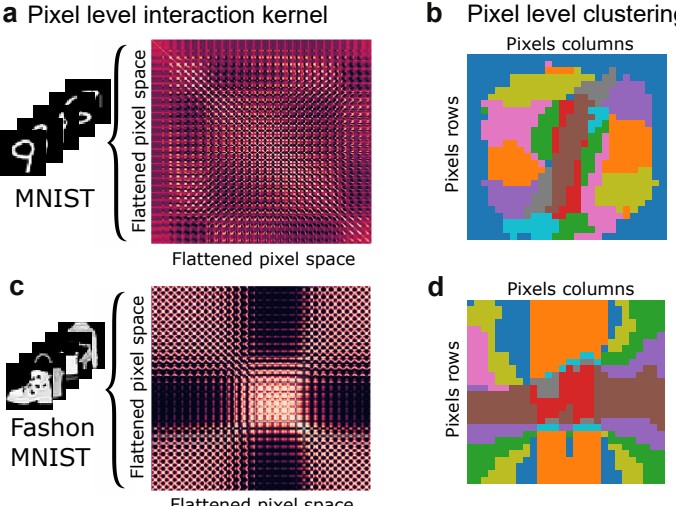

**Figure 4: Unsupervised encoding of Images.** Unsupervised learning of image embeddings as finite-support distributions (i.e., histograms) of pixel intensities. For every pixel position we assign a point location on the unit hypersphere and optimize such locations via the covariance operator *dataset embedding* w.r.t. the MDKE objective. (a) Samples from the MNIST dataset and learned pixel-to-pixel interaction kernel Gram matrix. (b) Spectral clustering of pixels based on the learned kernel Gram matrix. (c) and (d) same as (a) and (b) for Fashion-MNIST dataset.

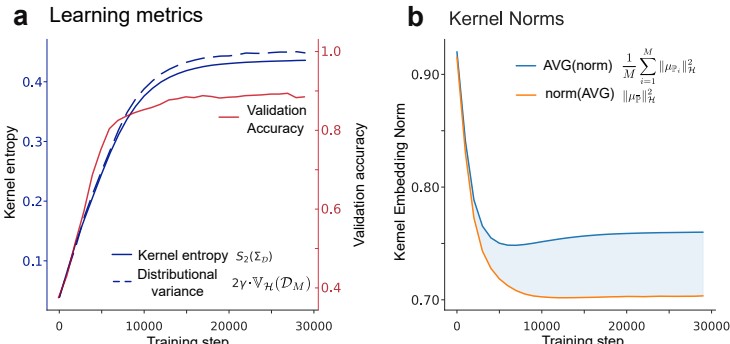

**Figure 5: Unsupervised encoding of Text.** Unsupervised learning of sentences embeddings as empirical distributions of words on the '20 Newsgroup' dataset. Goodness of the learned embeddings is evaluated by performing sentence-to-topic classification. (a) Distribution kernel entropy, distributional variance, and validation accuracy throughout training. (b) Kernel norms Eq. (16) throughout training. Shaded blue area (the difference between the blue and red lines) corresponds to the blue dotted line in panel (a) (up to a multiplicative factor).

## D.2 Text Classification Tasks

During the unsupervised pre-training phase, we observed a steady decrease in training loss (Fig. 5a) despite the small batch size (50 sentences with a total of $1,000$ words). Importantly, for this experiment we employed a regularized version of the objective (MDKE-R). We discuss the rationale and empirical evidence supporting the use of this regularization scheme in Appendix B.2. During evaluation, to reduce computational complexity, we sample $2,000$ sentences for the train and $1,000$ for the test split, keeping the classes balanced. The maximum classification accuracy achieved was approximately $89.3\%$, while the random initialization performance averaged at $37.5\%$ (Fig. 5). The framework's high accuracy in downstream classification showcases its prowess in learning potent latent representations, even when dealing with input distributions with large finite support.

# E    Implementation Details

In this section, we provide an example that illustrates the implementation of the proposed method using the PyTorch framework. All experiments were performed on a single machine with 1 GPU and 6 CPUs.

Functions to compute distribution kernel Gram matrix:

```python
def pairwise_kernel(x, gamma1):
    B, T = x.size(0), x.size(1)
    X_unroll = x.reshape(B*T, -1)
    dist = (X_unroll[:, None, :] - X_unroll[None, :, :])**2
    dist = torch.sum(dist, dim=2)
    G = F.avg_pool2d(
        dist[None, : , :],
        kernel_size=(T, T),
        stride=(T, T)
    )
    return torch.exp(-(gamma1/2.) * G.squeeze(0))

def distribution_kernel_gram(x, gamma1, gamma2):
    Gxy = pairwise_kernel(x, gamma1=gamma1)
    Gx = Gy = torch.diag(Gxy)
    G = Gx[:, None] + Gy[None, :] - 2*Gxy
    K = torch.exp(-(gamma2/2.) * G)
    return Gxy, K
```

Distribution kernel entropy estimator and the MKDE loss:

```python
def distribution_kernel_entropy(K):
    B = K.size(0)
    Cov = (1/B) * K
    return -(Cov ** 2).sum().log2()

def mkde_loss(encoder, X, gamma1, gamma2):
    # X.shape is (n_distributions, n_samples, d_input)
    Z = encoder(X)
    # Z.shape is (n_distributions, n_samples, d_latent)
    _, K = distribution_kernel_gram(Z, gamma1, gamma2)
    # K.shape is (n_distributions, n_distributions)
    loss = -distribution_kernel_entropy(K)
    return loss
```

The code presented here follows the setup presented in Sec. 5 using Gaussian kernel both as embedding and as a distribution kernel. Other kernels could be used by adjusting implementation of both helper functions accordingly. When training on large datasets, Charlier et al. [10] might be used to avoid memory overflow in average pooling.

