# OpenReview forum: "Learning to Embed Distributions via Maximum Kernel Entropy"
_NeurIPS.cc/2024/Conference — NeurIPS 2024 poster_

### Official Review · Reviewer_Z862 · 2024-07-07

**Soundness:** 3
**Presentation:** 3
**Contribution:** 3
**Rating:** 6
**Confidence:** 4

**Summary:**

This paper focuses on distributional regression (classification) that carries out supervised learning over a collection of datasets, where one instance (subject to one label) is a dataset that can be considered a distinct empirical distribution. The paper proposed a novel learning objective, maximize quantum entropy, to find a suitable embedding function f_{\theta} for all datasets on a sphere. The proposed quantum entropy is a the lower bound on distributional variance, where optimizing over such objective encourages inner-distribution variance reduction and inter-distribution variance increase for the given kernels, which intuitively lead to well-separability among distributions. Empirical simulation shows the potential of such objective in distributional supervised learning.

**Strengths:**

1. The paper is well-written and well-organized, the mathematical notation is on point and easy to follow. I enjoyed reading the paper.
2. The theoretical results are concise and straightforward, which provides enough motivation and context for the proposed framework.

**Weaknesses:**

1. My major concern is about the validity of the operation depicted in line 302-303, where the unsupervised pretraining is done with the whole dataset rather than a subset. This seems to be a slight lack of rigor and I hope the author could address how would MDKE perform when the pretraining is only done on the training dataset.
2. The second weakness is regarding the performance of MDKE in actual regression, for more detailed question, please refer to questions. At this point, the potential use case of MDKE seems rather limited (Distributional classification).

**Questions:**

As I am not an expert in distributional regression, I am unsure about how important regression analysis in this subject is. However, I do not see any empirical evaluation on anything that is technically regression (compared to classification). Moreover, the motivation behind the quantum entropy, as suggested in Figure 2a, is more aligned with classification (separability) than regression. Maybe the author could make a remark on this particular subject (regression), or provide a simple set of simulations on actual regression?

**Limitations:**

Yes

---

> ### Author Rebuttal · Authors · 2024-08-07
>
> Dear Reviewer,
>
> Thank you very much for your thoughtful review.
>
> Regarding the topic of unsupervised learning datasets, the Leukemia experiment conducts unsupervised kernel learning on the entire set of available distributions without accessing the corresponding labels. In the experiments described in Appendix D, unsupervised kernel learning is performed on subsets of the data.
>
> To address your question about classification versus regression: as it is stated in the abstract, the primary goal of this work was to study **discriminative tasks**. While we believe that minimizing latent variance ultimately facilitates solving regression problems as well, the experimental verification of this hypothesis was not within the scope of the current work.

---

> > ### Comment · Reviewer_Z862 · 2024-08-10
> > **Response**
> >
> > Thank you for the response. I have no more questions as all my previous questions were rather minor. I will keep my score of 6.

---

### Official Review · Reviewer_pXGE · 2024-07-09

**Soundness:** 3
**Presentation:** 3
**Contribution:** 3
**Rating:** 6
**Confidence:** 4

**Summary:**

The authors consider the distribution regression problem. The authors propose to learn an embedding function (i.e., embedding support data points into a sphere) and leverages the kernel embedding (into RKHS) and kernel functions (within RKHS space). The authors propose an algorithmic approach to learn such embeddings via maximum kernel entropy in an unsupervised fashion. Empirically, the authors illustrate advantages of the learned kernel with SVM on Tissue and Leukemia classification.

**Strengths:**

+ The authors propose an interesting framework to learn a kernel based on kernel embedding (into RKHS), parameterized by embedding function (into a sphere) in an unsupervised fashion.
+ The proposed method is well motivated. The paper is well-organized, easy to follow, with a detailed description for the proposed framework.
+ The interpretation and illustration of main concepts of the proposed method is also a plus.
+ The proposed learned kernel with SVM performs well in experiments.

**Weaknesses:**

+ It seems the experiments are quite weak. The authors only evaluate on two small datasets (with N <=50 distributions?) and on a classification task only (?). It is better to evaluate on larger datasets (especially in our current era)! It is only a plus to illustrate it on a real distribution regression problem.
+ It is better to report time consumption together with performances.

+ Another point is that the embedding into a sphere is not well motivated (e.g., why sphere? but not other spaces? How is about low-dimensional embedding, as other criteria for embedding?)

**Questions:**

The submission is well organized, and easy to follow. The authors also demonstrate detailed interpretations for the proposed approach. The proposed learned kernel approach is interesting and has good empirical results for the distribution regression problem. However, it seems that the experiments are quite weak.
+ The authors only evaluate on two small datasets (with N <=50 distributions?) and on a classification task only (?). It is better to evaluate on several larger datasets (especially in our current era)! Additionally, it is quite unusual to learn an embedding from only 16 distributions (for Tissue) and 20 distributions (for Leukemia). Could the authors clarify why it only needs a few input distributions to learn such embedding? Or only the number of supports in the distributions matters here?
+ It is only a plus to illustrate it on a real distribution regression problem.
+ It is better to report time consumption together with performances, and compare to recent kernel approach for distributions, e.g., based on optimal transport geometry.
+ It is also better to discuss the parameterization for the embedding and the proposed algorithm to learn it? It seems that the learning algorithm is non-convex?
+ Please give a proof for results in line 157-158? What is the gradient of L_MDKE w.r.t. \theta? Please describe S_2(\Sigma_D) w.r.t. K_D? and does the result in Equ. (13) only hold for the covariance operator \Sigma_D or also its estimator \hat\Sigma_D? Please clarify them.

Some other concerns are as follows:

+ The kernel hyperparameter (in Section B.1) is important and interesting. Could you elaborate it rigorously for the experiments with few distributions?
+ Could the authors comment on the choice of sphere for the embedding? Why not other spaces? It is also better to use a trivial baseline, e.g., learn embedding by autoencoder separately, and then apply the kernel embedding (into RKHS) and kernel functions (within RKHS) to illustrate the importance of the learned embedding?
+ It may be interesting to link the distribution regression with optimal transport (OT) geometry besides sliced-Wasserstein kernel [35], e.g., tree-Wasserstein kernel (Le et al., NeurIPS'19), Sobolev transport kernel (Le et al., AISTATS'22)? and other geometry on distributions? Could the authors comment on it?
+ It seems the proposed approach may be also related to the Log-Hilbert-Schmidt metric between positive definite operators (Ha Quang et al., NIPS'14), especially on application setup? Could the authors comment on it?

Briefly, the proposed method is interesting, and the submission will be stronger in case the authors have a further care for the experiments, e.g., with larger datasets, real distribution regression task.

**Limitations:**

Yes, the authors have a discussion on the limitations.

---

> ### Author Rebuttal · Authors · 2024-08-07
>
> Dear Reviewer,
>
> Thank you very much for your thoughtful review.
>
> Before addressing your specific questions, we want to emphasize that the leukemia diagnosis dataset used in the main part of our paper is a real dataset. It was collected during clinical studies, is sufficiently diverse, and represents the typical medical complexities involved in diagnosis. It has been published to facilitate research in early cancer diagnosis and treatment. From ML perspective, this dataset showcases typical problems in such tasks: large sample size (in millions of vectors), non-i.i.d sampling due to a small number of subjects, and challenges in normalizing vector representations because of the nature of the measurements (e.g., different biomarkers). Appendix D includes experiments with more common ML modalities to demonstrate the applicability of our framework.
>
> To address your questions:
>
> * Regarding the question of using only a few input distributions. Specifically for the Leukemia dataset, the input sample space is continuous, and the encoder function is a non-linear map (i.e. NN). This complexity makes it difficult to estimate the number of distributions required to learn an encoder that sufficiently separates parts of the support space for downstream tasks. While such a quantitative statement would be beneficial for practical applications, we currently lack these estimates, even for much simpler setups.
> * Regarding time consumption, we report in the paper the computational resources required for our experiments as not significant.
> * The non-convexity of the objective is discussed alongside our choice of optimization algorithm in lines 157-163.
> * The learning objective involves the logarithm of the squared Frobenius norm of the kernel Gram matrix, and the gradient can be easily obtained using any autodiff software. Given that the encoder function is a neural network, deriving the gradient manually may be tedious, and we believe it does not provide additional theoretical value for the discussion in our paper. Computational form of $S_2(\Sigma_D)$ w.r.t. $K_D$ is given in Eq. 14 on page 5. Eq. 13 is stated in terms of true operator $\Sigma_D$, the rest relies on the fact that $\hat{\Sigma}_D$ is a consistent estimator of $\Sigma_D$ (with the proof of this fact given in [1]).
> * The hyperparameter selection procedure we applied is defined and analyzed in [2].
> * Regarding the choice of a hypersphere as the latent space: as we stated in lines 141-143, the latent space is a design choice. Other compact spaces with sufficient symmetries and the existence of proper kernels (see Assumption 3.1) would also be valid. The hypersphere is a simple and commonly used latent encoding space in ML, possessing the properties required.
> * While we acknowledge that a specialized kernel suited to the task's geometry might improve solution quality, the study aims to learn the kernel from the dataset (or, more broadly, identify what makes a kernel "good" for solving discriminative tasks on distributions).
> * Currently, we do not see an apparent connection to the Log-Hilbert-Schmidt metric. Note that in our work, we do not use covariance operators to define the kernel. The operator embedding is only defined as a learning objective. Once learning is complete, the resulting kernel does not depend on the covariance operators.
>
> [1] Bach, Francis. "Information theory with kernel methods." IEEE Transactions on Information Theory 69.2 (2022): 752-775.
>
> [2] G. Blanchard, G. Lee, and C. Scott. Generalizing from several related classification tasks to new unlabeled sample. Advances in neural information processing systems, 24, 2011

---

> > ### Comment · Reviewer_pXGE · 2024-08-10
> >
> > I thank the authors for the explanation in the rebuttal.
> >
> > I have some quick questions as follows:
> >
> > **(1)Time consumption**
> > - It is not clear to me why the mini-batch SGD is efficient for the proposed non-convex learning problem yet? There is no analysis for it (e.g., convergence, affects of initialization, stopping condition, step-size, etc)? Could you elaborate why the computational time is not significant?
> > - I agree that in Section 5, the authors show the advantages on performances for the proposed method. However, it is not clear about other aspects, e.g., algorithms to solve the non-convex learning problems?
> >
> > **(2) Data-size is too small**
> > - I agree that the used datasets are real-world.
> > - However, their size is just too small! The applications use kernel SVM (i.e., non-linear classification), how many samples do the authors use for training such classifiers (split 70/30 from a very small datasets with less than 50 samples?) Could the authors elaborate it more rigorously.
> >
> > **(3) Gradient formulation (for minibatch SGD)**
> > - the gradient is essentially the most important part for SGD approach.
> > - How does the usage of autodiff affect the convergence and learning procedure?
> > - Is it possible to derive the formulation of the gradient?

---

> > > ### Author Response · Authors · 2024-08-12
> > >
> > > Thanks for engaging on technical details. We really appreciate and enjoy discussing this with you:
> > >
> > > **(1) Time consumption**
> > >
> > > Mini-batch SGD is a common approach for tackling non-convex problems, typically able to overcome local minima due to the stochasticity of gradient estimates. Since it is usually not feasible to predict the performance of an SGD solution for a specific problem at hand, we propose it as an efficient method for solving the described optimization objective based on empirical evaluations. Note that the objective is convex w.r.t. to the kernel Gram matrix. The use of a neural network as the encoder turns it non-convex w.r.t. to parametrization of the NN. However, in our experience, SGD remains a suitable choice, often yielding satisfactory results in such scenarios. While we obviously recognize the value of quantitative statistical guarantees, our implementation of SGD aligns with standard practices in the literature for deep NNs and we expect the properties of our implementation to be mostly inherited by studies of mini-batch SGD convergence in deep-NNs.
> > >
> > > Regarding computational time, our process is relatively efficient, typically requiring only hours on a single machine. This negates the need for extensive parallelization across multiple GPUs or machines, underscoring our method’s flexibility and accessibility.
> > >
> > > **(2) Data-size is too small**
> > >
> > > We tried to explain at best the dataset but, as for any dataset, real insight comes only by playing with it. This is actually not a very small dataset: it has millions of samples! Each distribution has 10^5-10^6 samples on which the optimization is performed. What keeps the dataset compact is the dimension of each sample ranging from 20 to 50 based on the selected criteria. While the dataset’s dimensionality is smaller than what you might find in image datasets, the number of samples—which we refer to as the “distribution part”—is quite extensive. The apparent simplicity of the downstream classification task we use to properly assess quality of the distribution kernel learned is the result of specific modeling and parameterization choices made. We hope it demonstrates the power of the techniques applied.
> > >
> > > We use 70/30 splits for training, details are provided in the opening paragraphs of the Section 5.
> > >
> > > **(3) Gradient formulation (for minibatch SGD)**
> > >
> > > We agree with the Reviewer that the gradient is the most important building block of the learning procedure. We do not expect any substantial improvement from replacing autodiff with a theoretically derived custom gradient iteration. Moreover, by establishing a general framework, we aim to keep it flexible: any differentiable encoder can be incorporated without changes to the outer training loop, leveraging the power of modern autodiff software. We acknowledge that we might be wrong in such intuition and we are curious to know if the Reviewer suspects a different outcome. However, for this reason, while it is possible to manually derive a custom gradient, we didn't devote too much time to it.

---

### Official Review · Reviewer_m5U2 · 2024-07-10

**Soundness:** 2
**Presentation:** 3
**Contribution:** 2
**Rating:** 6
**Confidence:** 3

**Summary:**

This work studies the distribution regression problem. The inputs are distributions, and the goal is to learn embeddings for these inputs. The authors propose the following method: First map the distributions to the kernel mean embeddings with respect to a embedding kernel $k_{emb}$, and then conduct a kernel regression on a distribution kernel $K_{distr}$. The authors propose to learn $k_{emb}$ by maximizing the quantum entropy. The effect of this is minimizing the variance within each distribution, while maximizing the spread of distributions.

**Strengths:**

**Note:** I reviewed this paper at ICML 2024. This review is also based on the difference between the two versions.

1. This paper studies an important problem, which I believe has potential applications in many domains. The paper is well written and easy to read. The intuition behind the proposed method is clearly demonstrated. The proposed approach, as far as I know, is new. Thus, this paper contributes valuable insights and methodologies to the literature.

2. The geometric interpretation, that is minimizing the variance and maximizing the spread, makes sense and is similar to some related methods such as contrastive learning (see Wang & Isola, 2020).

3. The paper is well-structured, offering a comprehensive and clear exposition of the background, methodology, and theoretical aspects of the proposed approach. This clarity enhances the paper's accessibility to readers who might not be intimately familiar with the nuances of kernel methods or distributional regression, thereby broadening its potential impact.

**Weaknesses:**

Several weaknesses were discussed at the ICML review, which are not fully addressed in the new version:

1. The major weakness is the experiment part, which only considers the flow cytometry task. While this might be an important application (which I am not familiar with), the usefulness of the proposed method in applications is still questionable. Experiments on images and text are moved to Appendix D, while they were in the main body in the previous version. For me I would prefer them to be in the main body. As I pointed out last time: "One thing that the authors could do to increase the impact of this work is to apply this method to more real tasks. For instance, the authors mentioned voting behavior prediction and dark matter halo mass learning in the introduction, but these data sets are not used in the experiments."

2. Other weaknesses in the empirical part pointed out by other reviewers:
  - Analysis of Important Variables: The experimental setup lacks a thorough analysis of crucial variables, and does not have ablation studies. The impact of hyperparameter selection on the method's performance should be studied and discussed in the paper.
  - Computational Cost and Scalability: The authors acknowledge this point in the limitations.
  - Reproducibility: The authors promise to release their code in the main text.

3. One reviewer asked last time why not directly optimizing $V_H$, which is an upper bound of the proposed objective as proved in Eqn. (17). The authors explained that directly optimizing $V_H$ has a lower performance, but did not provide any experiment results. I am not satisfied with this explanation. I suggest the authors add a direct comparison in the paper. This could be an important ablation study.

**Conclusion:** Overall, I am still in favor of accepting this paper. This paper indeed has lots of issues, but it is a preliminary study of an important problem and has some interesting insights, which I believe could inspire future work. My current rating is weak accept, and I am willing to raise it to accept based on the authors' response.

**Questions:**

See weaknesses.

**Limitations:**

Limitations are discussed

---

> ### Author Rebuttal · Authors · 2024-08-06
>
> Dear Reviewer,
>
> We are grateful for your meticulous review and constructive feedback.
>
> Since our initial submission to ICML, the manuscript has undergone significant improvements, particularly in the experimental section, prompted by critiques akin to those presented by the Reviewer.
>
> We concur that incorporating additional experiments within the main text could be advantageous. However, we opted to relegate the image and text modalities to the Appendix to mitigate potential confusion.
> The feedback we received indicated a prevalent misunderstanding among readers. They often speculated about the relevance of supervised methods, which are standard for the datasets in question (e.g. MNIST), to our research framework. However, our framework is unsupervised and the quasi entirity of the suggested methods are not comparable to our framework. Therefore, we chose to highlight an experiment where the application of a distributional regression model is more evident.
> The leukemia diagnosis dataset, derived from clinical trials, best represents the complexities inherent in medical diagnosis and illustrates typical machine learning challenges, such as large sample sizes and non-i.i.d. sampling.
>
> Regarding the other points of critique, we recognize that scalability is a challenge for the practical deployment of kernel methods. We intend to conduct a more thorough analysis of the hyperparameter selection process. Nevertheless, we have shown that our method remains viable for small to medium-sized datasets, where scalability issues are less pronounced. The principal contribution of our paper is the establishment of a theoretically sound objective for unsupervised learning of a data-dependent distributional kernel. This contribution is innovative both as a theoretical construct and in its implications for the application of kernel-based methods to discriminative tasks involving distributions. We anticipate that this novel perspective will stimulate further interest and research in the field. We welcome efforts by other research groups to adapt and scale our methods for use with larger datasets.
>
>
> A brief comment about using $\mathbb{V}_\mathcal{H}$ as an optimization objective. We will incorporate an ablation study in the experimental section. Although the outcomes do not surpass those of a random encoder, which is expected as per theoretical considerations discussed in the paper.

---

> > ### Comment · Reviewer_m5U2 · 2024-08-09
> > **Response**
> >
> > I thank the authors for the response. I don't have any further questions at this point. I will discuss with my fellow reviewers and the AC, and notify the authors if I change my rating.

---

### Official Review · Reviewer_bRTz · 2024-07-13

**Soundness:** 4
**Presentation:** 3
**Contribution:** 2
**Rating:** 4
**Confidence:** 4

**Summary:**

The authors propose a method for learning kernel embeddings for distributions via maximizing a Renyi entropy objective. They relate their objective to the distributional variance of the embeddings, and explain why this can lead to good embeddings for downstream tasks. Empirical evaluations show good results on a flow cytometry dataset, but results on image and text classification are weaker.

**Strengths:**

- This paper is very clearly written, with good explanations on kernel distribution embedding and covariance operators, and the two-stage embedding scheme used in the estimation. There are also good motivation on the use of entropy maximization as an objective for learning good feature embeddings.

- The authors provide theoretical analysis on the relation between second-order Renyi entropy and distribution variance, and how maximizing the entropy helps increase the variance of the feature embeddings.

**Weaknesses:**

- The weaker part of this paper is the empirical evaluations. There is only one dataset provided in the main paper on flow cytometry which the proposed method gives improvements over existing methods. There are small-scale experiments on image and text classification in the Appendix. However, even as the results of the learned embeddings are better than randomly initialized embeddings, they are not as good as direct SGD optimization using a cross-entropy loss on MNIST or 20 News-groups. This cast doubt on the effectiveness of the entropy maximization objective for feature learning in text and image problems.

- Another weakness of the current method is scalability. From Equation 14 the objective scales quadratically with the number of distributions. This makes the computation expensive and indeed most of the experiments described in this paper are rather small-scale.

**Questions:**

- How is the sampling done in the mini-batch SGD? There are two levels in the two-stage kernel embedding process, one with the individual samples and one with the distributions. How do the authors pick the SGD samples to ensure good optimization progress? And what are the required batch sizes relative to the training dataset size?

**Limitations:**

Yes

---

> ### Author Rebuttal · Authors · 2024-08-06
>
> Dear Reviewer,
>
> Thank you very much for your thoughtful review.
>
> Upon reading the Reviewer's comments we were confused by what exactly the Reviewer contends when referring to embeddings learned by SGD via Cross-Entropy (CE). Our methodology is inherently unsupervised, whereas CE intrinsically necessitates label information. Consequently, juxtaposing our approach with SGD optimization of the CE loss does not constitute a congruent comparison, given that our optimization goal is unsupervised—learning a data-dependent kernel between distributions devoid of label access—unlike the supervised nature of CE. We invite any additional insights from the Reviewer should there be an alternate perspective.
>
>
> On the other hand, we agree that our method’s scalability to larger datasets warrants further work. However, we posit that scalability challenges do not preclude the applicability of our method to practical datasets. Our methodology has demonstrated superior performance in real-world contexts, such as clinical studies. The leukemia diagnosis dataset featured in our study, derived from clinical trials, is representative of the complexity typically encountered in medical diagnoses and has been made available to support research in early cancer detection and therapy. This dataset exemplifies common challenges in the same class of ML applications:
> a large sample size (in millions of vectors), non-i.i.d. sampling due to a small number of subjects, and challenges in normalizing vector representations because of the nature of the measurements (e.g., different biomarkers). It is our stance that the utility of methods like ours should not be contingent solely on their suitability for the largest available datasets. Our approach is equally applicable to small and medium-sized datasets, as evidenced by our findings.
>
>
>
> To address specific questions about the sampling procedure, both levels of sampling are performed uniformly to obtain a batch of i.i.d. samples of a given size. We do not have quantitative guarantees for the optimal batch size, and the best hyperparameters were determined using a standard grid search. In addition, in the experiments conducted for this study, our empirical observations suggest that batch size is not a critical determinant of the downstream performance.

---

> > ### Comment · Reviewer_bRTz · 2024-08-09
> >
> > Yes, when I say SGD optimization with CE loss I do mean supervised learning with labels. In computer vision there is a line of work on unsupervised representation learning with contrastive methods such as SimCLR or MoCo, and the performance of these methods is on par with supervised learning on ImageNet. So it is not completely unfair to compare the performance of unsupervised representation learning with their supervised counterpart. The same is true in NLP, where unsupervised representation learning (although via pretraining on large amount of texts) beats supervised learning with limited data. It could be difficult to require a general unsupervised learning method like the author's proposal to do better than domain-specific approaches used in vision or NLP, but it also limits the applicability of the proposed algorithm on these type of data since there are strong alternatives.

---

> > > ### Author Response · Authors · 2024-08-09
> > >
> > > Thank you for your valuable feedback. Your clarification has greatly helped us understand your perspective.
> > >
> > > We recognize that your emphasis on state-of-the-art (SOTA) performance is aimed at advancing current methodologies. However, we believe this focus may overlook the foundational nature of our contribution. Our work seeks to establish a new approach in unsupervised kernel learning, which we believe holds significant potential for future research and development. If we understand correctly, you are requesting a comparison between unsupervised and self-supervised approaches (e.g., SimCLR, MoCo) within a domain-free setup like ours.
> > >
> > > This is indeed a fundamental question, but one that other subfields (such as vision and language) have addressed at more mature stages. Our contribution is the first to propose unsupervised learning of full distribution embeddings. While we appreciate your feedback and find it very helpful, we feel that rejecting our contribution based on these remarks may be asking too much from a single paper.
> > >
> > > Here are some specific comments on your remarks:
> > >
> > > 1. **Domain-Specific vs. Domain-Free Approaches**: Our algorithm is designed to be domain-free, which we believe offers a significant advantage over domain-specific self-supervised methods. This flexibility allows our approach to be applied across various domains without requiring domain-specific adjustments.
> > >
> > > 2. **State-of-the-Art (SOTA) Focus**: While we acknowledge the importance of achieving SOTA results, our primary goal in this paper is to introduce a novel framework for unsupervised learning of whole distribution embeddings. We believe that establishing this new framework is a crucial first step that can pave the way for future improvements and comparisons.
> > >
> > > Thank you once again for your insightful comments. We look forward to addressing them in our future work.

---

### Official Review · Reviewer_1Qk4 · 2024-07-24

**Soundness:** 3
**Presentation:** 4
**Contribution:** 2
**Rating:** 6
**Confidence:** 3

**Summary:**

This paper proposes a new unsupervised way to learn kernels for distribution regression through entropy maximization. In addition, they also propose a geometric interpretation which is very interesting. The learnt kernels are general and have shown on some experimental settings to perform better than standard unlearnt kernels.

**Strengths:**

- The paper proposes a novel way to learn kernels through entropy maximiazation in the setting of distribution regression.
- The paper is very well written and easy to follow for someone who has worked the KME before.
- The paper shows promising results compared to exisiting methods that do not train the kernel (through entropy)

**Weaknesses:**

- The experimental section does not compare to other kernel distribution regression methods. Could the authors confirm why [1] was omitted in the comparisons?
- In addition citations to other methods that also learn kernel albeit not with entropy should be cited [2, 3, 4, 5]
- How were the hyper parameters picked for the fixed kernel such as RBF kernel? Can you please lay out the whole process as well as the corresponding values for each of the hyper parameters?



[1] Bayesian approaches to distribution regression
[2] Learning Deep Features in Instrumental Variable Regression
[3] Noise Contrastive Meta-Learning for Conditional Density Estimation using Kernel Mean Embeddings
[4] Meta Learning for Causal Direction
[5] Deep proxy causal learning and its application to confounded bandit policy evaluation

**Questions:**

see above

**Limitations:**

yes

---

> ### Author Rebuttal · Authors · 2024-08-06
>
> Dear Reviewer,
>
> Thank you very much for your thoughtful review.
>
> * We appreciate the insightful comments from the Reviewer. We will incorporate the Bayesian approaches they highlighted into our extended discussion in Appendix C. We have dedicated this appendix to a thorough review of related literature. The absence of a comparison with the work of [1] in our experimental section is due to the novel nature of our task, which is the unsupervised learning of distribution embeddings. To our knowledge, [1] and similar studies leverage labels in a supervised manner, which contrasts with our unsupervised approach. A direct comparison would necessitate significant modifications to these methods, constituting a separate contribution.
> * In response to the references suggested by the reviewer, we will enrich our manuscript with additional citations and a succinct discussion of [2,3,4,5].
> * We had delineated our hyperparameter selection process in Section B.1 in detail. To enhance clarity, we will insert a second cross-reference to this section. This procedure builds upon and elaborates the methodology established in [6].
>
> [1] Bayesian approaches to distribution regression
> [2] Learning Deep Features in Instrumental Variable Regression
> [3] Noise Contrastive Meta-Learning for Conditional Density Estimation using Kernel Mean Embeddings
> [4] Meta Learning for Causal Direction
> [5] Deep proxy causal learning and its application to confounded bandit policy evaluation
> [6] G. Blanchard, G. Lee, and C. Scott. Generalizing from several related classification tasks to a new unlabeled sample. Advances in Neural Information Processing Systems, 24, 2011.

---

> > ### Comment · Reviewer_1Qk4 · 2024-08-08
> > **response**
> >
> > I thank the authors for the clarifications and i will keep my score of 6.

---

### Decision · Program_Chairs · 2024-09-25

**Decision:**

Accept (poster)

**Comment:**

This paper addresses the kernel learning problem with the maximum kernel entropy, which is one of the most important (and often neglected) problems in distribution regression. The reviewers unanimously concur on the importance of the problem and the novelty and suitability of the proposed methods. Nevertheless, the reviewers also pointed out several limitations (e.g., small datasets, marginal improvement, etc) in the empirical evaluations of the proposed methods. I encourage the authors to further address these concerns in the camera-ready version of the manuscript, for example, by clarifying the experimental setup and moving some of the results from the appendix to the main paper.